# ReFusion: A Diffusion Large Language Model with Parallel Autoregressive Decoding

**Jia-Nan Li**[1, 2, 3, 4*]   **Jian Guan**[4*]   **Wei Wu**[5†]   **Chongxuan Li**[1, 2, 3†]

[1]Gaoling School of Artificial Intelligence, Renmin University of China
[2]Beijing Key Laboratory of Research on Large Models and Intelligent Governance
[3]Engineering Research Center of Next-Generation Intelligent Search and Recommendation, MOE
[4]Ant Group   [5]Ant International
{lijianan, chongxuanli}@ruc.edu.cn
{jianguanthu, wuwei19850318}@gmail.com

 **Code:** https://github.com/ML-GSAI/ReFusion
 **Model:** https://huggingface.co/GSAI-ML/ReFusion

## Abstract

Autoregressive models (ARMs) are hindered by slow sequential inference. While masked diffusion models (MDMs) offer a parallel alternative, they suffer from critical drawbacks: high computational overhead from precluding Key-Value (KV) caching, and incoherent generation arising from learning dependencies over an intractable space of token combinations. To address these limitations, we introduce ReFusion, a novel masked diffusion model that integrates sequence reorganization into the causal attention framework. By elevating parallel decoding from the token level to a higher slot level, ReFusion interleaves inter-slot diffusion-based selection with intra-slot autoregressive infilling, while reordering newly generated slots ahead of the remaining masks after each iteration. Consequently, this design simultaneously unlocks full KV cache reuse and reduces learning complexity from an intractable token combination space to a manageable slot-level permutation space. Extensive experiments on seven diverse benchmarks show that ReFusion not only overwhelmingly surpasses prior MDMs with a 34% performance gain and an over $18\times$ speedup on average, but also bridges the performance gap to strong ARMs while maintaining a $2.33\times$ average speedup.

## 1 Introduction

While autoregressive models (ARMs) (Grattafiori et al., 2024; Yang et al., 2025; Jaech et al., 2024) have achieved remarkable progress across a wide range of tasks (Chen et al., 2021; Wei et al., 2022; Lightman et al., 2023; Li et al., 2024), their inference throughput is fundamentally limited by a sequential, left-to-right decoding process that precludes parallelization (Chen et al., 2023; Cai et al., 2024; Zhang et al., 2025). In contrast, masked diffusion models (MDMs) (Nie et al., 2025; Ye et al., 2025) operate via an iterative denoising process with no fixed generation order. This flexibility yields two significant advantages. First, it permits parallel decoding by assuming conditional independence among target tokens: their joint probability, given the context, is assumed to be the product of their individual marginal probabilities (Li et al., 2023). Second, it offers the potential for the model to discover better generation orders than the rigid left-to-right trajectory (Kim et al., 2025).

Despite these theoretical advantages, existing MDMs often suffer from two issues: **(1) Architectural bottlenecks negate efficiency gains from parallelism.** The flexibility of generation orders requires bidirectional attention in MDMs (Vaswani et al., 2017; Devlin et al., 2019), an architectural choice fundamentally incompatible with Key-Value (KV) caching used in ARMs (Radford et al., 2018). That is, each decoding iteration forces a full re-computation of the KV states of the entire context, introducing substantial latency and making MDMs significantly slower than ARMs (Feng et al.,

---

*Equal contribution.
†Corresponding authors: Wei Wu and Chongxuan Li.

2025). **(2) Intractable learning complexity hinders coherent parallel generation.** MDMs typically decode multiple tokens with high marginal probabilities in parallel (Nie et al., 2025). However, the conditional independence assumption frequently fails for these tokens, particularly for nearby tokens, leading to severe incoherence (Huang et al., 2022; Luxembourg et al., 2025; Gwak et al., 2025). For example, in a context where both "at once" and "right now" are valid, an MDM might decode a spurious output "right once" by independently sampling tokens with high individual marginal probabilities but low joint probability. We attribute this failure to an immense learning challenge: modeling a data distribution over an exponential space of possible token combinations is far more demanding than the fixed sequential dependency of ARMs. Consequently, current MDMs often remain undertrained for reliably identifying conditionally independent tokens.

To address these challenges, we introduce REFUSION, a masked diffusion large language model leveraging sequence reorganization with a causal attention framework. Specifically, we partition the masked sequence into fixed-length, consecutive sub-sequences, referred to as slots. During each decoding step, REFUSION performs diffusion-based slot selection followed by parallel autoregressive infilling within these slots. Subsequently, the newly decoded slots are reordered to the front of the remaining masked slots (Figure 3). This design yields two critical benefits: **(1) Full KV Cache Reuse:** REFUSION seamlessly reuses the KV states of *all* previously generated tokens without sacrificing the MDM's flexible generation order. **(2) Reduced Learning Complexity:** By serializing adjacent tokens within a slot, REFUSION mitigates the conditional independence violations typical of MDMs. This significantly reduces the learning complexity, transforming an intractable token combination space into a manageable slot permutation space. Appendix A.1 compares REFUSION and existing MDMs in detail.

REFUSION's training mirrors its inference dynamics. For each sequence, we randomly mask several slots, permute the clean slots, and reorder the input so that clean slots precede masked ones. The model is then optimized with a hybrid objective that cultivates its dual capabilities: an autoregressive loss on the permuted clean slots for sequential generation, and a denoising loss on the masked slots for context-aware parallel reconstruction. Unlike traditional MDMs, which learn only from masked positions, this hybrid objective supervises every token, boosting data efficiency.

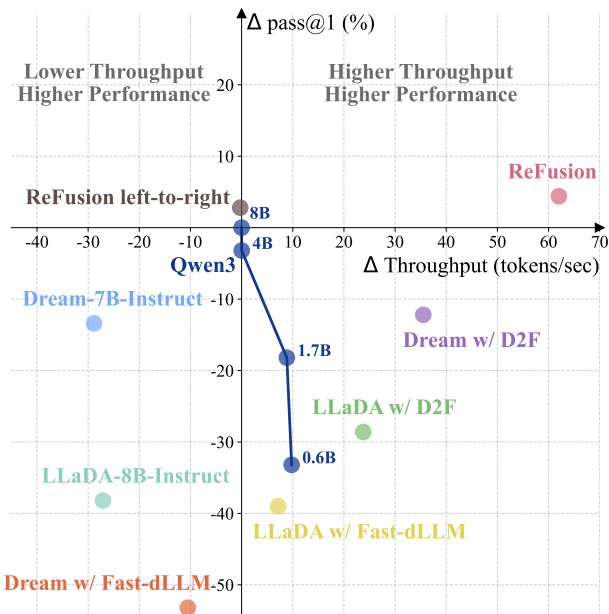

Our extensive experiments on seven benchmarks spanning math, code generation, and general-purpose understanding and reasoning demonstrate that REFUSION decisively establishes a new state-of-the-art for MDMs. Compared to LLaDA (Nie et al., 2025) and Dream (Ye et al., 2025), RE-FUSION achieves an average performance gain of 34% while being over 18× faster in throughput (tokens/sec). More strikingly, REFUSION consis-

Figure 1: Performance-throughput trade-off on MBPP. We plot pass@1 (%) against throughput (tokens/sec), with both metrics calculated relative to the Qwen3-8B baseline at the origin. The "REFUSION left-to-right" ablation forces serial decoding using the REFUSION model.

tently challenges and often surpasses strong ARMs. For instance, it outperforms Qwen3-8B (Yang et al., 2025) on GSM8K (Cobbe et al., 2021) and MBPP (Austin et al., 2021) by 3.68 absolute points while being 2.33× faster on average. This dual advantage in both performance and speed is further illustrated in Figure 1, where REFUSION (the red point) pushes the performance-efficiency boundary significantly towards the top-right quadrant. It significantly outperforms both the Qwen3-series (the blue line) and prior MDM-based methods, which are situated in the bottom-left or bottom-right regions, indicating that they lag behind in throughput, performance, or both dimensions. Furthermore,

our controlled experiments confirm that these gains are driven by our architectural and training innovations, rather than initialization or data advantages.

Our contributions are summarized as follows:

• We propose REFUSION, a generative model integrating inter-slot parallel decoding with intra-slot autoregressive decoding, combining the strengths of autoregREssive and difFUSION-based modeling.

• To the best of our knowledge, REFUSION is the first MDM that achieves full KV cache reuse of every decoded token, while maintaining global generation flexibility and tractable learning complexity.

• Extensive experiments on seven diverse benchmarks show that REFUSION not only overwhelmingly surpasses all prior MDMs in both performance and speed, but also bridges the performance gap to ARMs while maintaining the efficiency advantage.

## 2 RELATED WORK

MDMs promise to outperform traditional ARMs by offering faster inference through parallel decoding and potentially superior solutions via flexible generation orders (Kim et al., 2025). Recent MDMs such as LLaDA (Nie et al., 2025), the first open-source MDM trained from scratch, and Dream (Ye et al., 2025), initialized from an ARM, have delivered performance on par with ARMs of equivalent scale across diverse tasks, establishing MDMs as a viable research direction.

**Architectural Designs for Efficient MDMs.** Standard MDMs' reliance on bidirectional attention precludes the use of KV caching. Recent work alleviates this bottleneck through three main strategies. The first strategy approximates KV cache reuse while retaining bidirectional attention. For instance, dLLM-Cache (Liu et al., 2025) reuses slow-changing KV states, while sparse-dLLM (Song et al., 2025) dynamically prunes non-critical KV states. The second strategy mixes bidirectional attention and causal attention. Models like BD3-LMs (Arriola et al., 2025) and Fast-dLLM (Wu et al., 2025a) partition the sequence into consecutive blocks, enforcing a left-to-right order between blocks to enable KV cache reuse, while retaining parallel, bidirectional generation within each block. D2F (Wang et al., 2025) further parallelizes the generation of succeeding blocks, although performance is limited by the lack of inter-block lookahead attention. The final strategy leverages only causal attention, enabling an exact KV cache. Eso-LMs (Sahoo et al., 2025), for instance, dynamically reposition newly generated tokens ahead of masked ones at each step to facilitate caching. However, this strategy introduces an intractable learning objective at a token-level permutation space, which hinders training and leads to significant performance drops.

**Decoding Strategies in MDMs.** A crucial aspect of MDM inference is the strategy used to select which tokens to decode in parallel at each step. Existing approaches generally fall into two categories. The first class leverages confidence heuristics derived from the model's own distribution, such as top token probability (Nie et al., 2025), low entropy (Ben-Hamu et al., 2025), and probability margins between top candidates (Kim et al., 2025). Some methods further refine these heuristics with position-aware weights and frequency-based calibration (Huang et al., 2025). While simple, these methods rely on the often-unreliable assumption that the model's confidence scores are perfectly calibrated (Wu et al., 2025a). The second class employs external models for verification, e.g., using a small ARM to validate and extend the longest acceptable prefix (Hu et al., 2025; Israel et al., 2025), or using dedicated reward models to guide generation (Gwak et al., 2025). Although effective, these approaches introduce the overhead of maintaining and querying a separate model. Unlike these methods, REFUSION adopts a unified inference framework that benefits from the parallel efficiency of MDMs without sacrificing the quality assurance of ARMs, all within a single architecture.

**Comparison with Block-based Diffusion.** Notably, while REFUSION shares the concept of grouped processing units with block-based diffusion methods (Arriola et al., 2025), our *slot* design is fundamentally distinct from the *block* across the following dimensions:

• **Motivation:** Slots are designed to reduce learning complexity to ensure coherent generation, whereas blocks are primarily introduced to enable KV cache reuse.

• **Operation:** Slots are parallel inter-slot and serial intra-slot. Blocks are exactly the opposite: serial inter-block and parallel intra-block.

• **Characteristics:** Slots in REFUSION support full KV cache reuse while maintaining global generation flexibility. Block-based methods, however, sacrifice this flexibility for a left-to-right inter-block schedule, and intra-block bidirectional attention precludes KV caching and risks incoherence.

• **Compatibility:** Slots are hierarchically compatible with blocks, as they can be nested within each block, positioning REFUSION as a more generalizable framework.

## 3 PRELIMINARY

**Autoregressive Models.** ARMs are a prominent class of generative models that factorize the joint probability of a sequence $x = (x_1, \ldots, x_L)$ by enforcing a strict left-to-right conditional dependency using a causal attention mask. This structure leads to a next-token prediction objective, where the model parameters $\theta$ are optimized by minimizing the negative log-likelihood: $-\sum_{i=2}^{L} \log P_\theta(x_i \mid x_{<i})$. During inference, generation is an inherently sequential process requiring $T$ forward passes to produce a sequence of length $T$, resulting in latency that scales with the sequence length.

**Masked Diffusion Models.** MDMs represent another class of generative models that operate on a "mask-and-denoise" principle. During training, each sample $x_0 = (x_0^1, x_0^2, \cdots, x_0^L)$ is corrupted to $x_t$ by masking each token with a special token "`[MASK]`" with probability $t \sim U(0, 1)$. The model learns to reconstruct the original sequence by minimizing the objective: $-\frac{1}{t} \sum_{i=1}^{L} \mathbf{1}(x_t^i = $ `[MASK]`$) \log P_\theta\left(x_0^i \mid x_t\right)$. MDM inference proceeds by progressively generating tokens from a fully masked sequence. It requires fewer forward passes than an ARM thanks to parallel decoding, but each pass is drastically more expensive due to its incompatibility with KV caching.

## 4 METHODOLOGY

### 4.1 SEQUENCE REORGANIZATION

To address the aforementioned limitations of MDMs, we propose two methods of sequence reorganization: token reorder to simultaneously support full KV caching and globally flexible decoding; and slot partition to reduce learning complexity for a more manageable objective.

**Token Reorder for Full KV Cache Reuse.** RE-FUSION adopts a standard causal attention similar to traditional ARMs, while performing global position-flexible decoding during inference as in MDMs. Specifically, REFUSION moves newly decoded tokens to the front of the remaining masked tokens

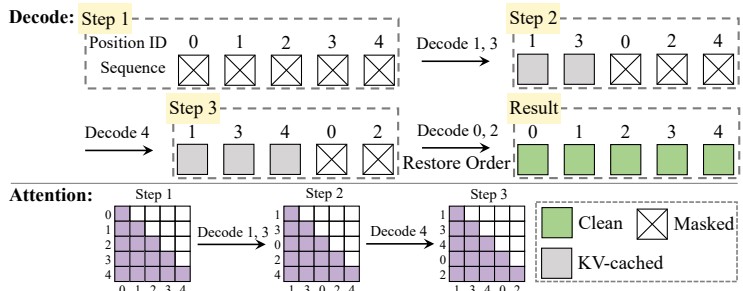

Figure 2: Illustration of full KV cache reuse via token reordering.

while preserving their relative internal order (Figure 2). Consequently, before each decoding step, all decoded tokens appear contiguously at the beginning, followed by the remaining masked positions. This layout enables the use of full KV cache at every decoding step.

However, this strategy introduces a mismatch between the input positions of decoded tokens and their original positions in the correct sequence, which may adversely affect attention computation and semantic modeling. To address this, we compute attention using position IDs that correspond to each token's original index, ensuring that attention scores remain invariant to token reorder during decoding.

Taking the widely used Rotary Position Embedding (RoPE) (Su et al., 2021) as an example, the attention computation between a query $\boldsymbol{q}_m$ with position ID $m$ and a key $\boldsymbol{k}_n$ with position ID $n$ is formulated as:

$$f(\boldsymbol{q}_m, \boldsymbol{k}_n) = (\boldsymbol{R}_m \boldsymbol{q}_m)^\top (\boldsymbol{R}_n \boldsymbol{k}_n) = \boldsymbol{q}_m^\top \boldsymbol{R}_{n-m} \boldsymbol{k}_n, \tag{1}$$

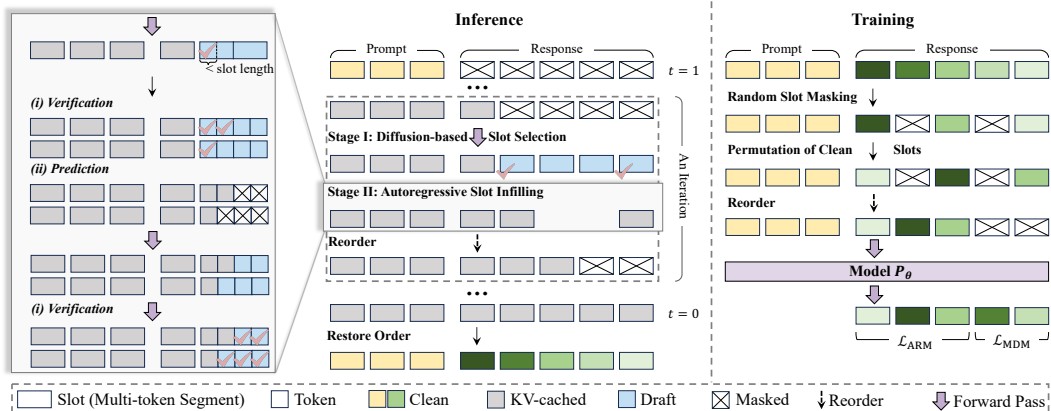

Figure 3: Overview of REFUSION. Left (Inference): An iterative slot-level "select-and-infill" loop. A diffusion stage selects and drafts slots, followed by parallel autoregressive verify-and-predict infilling. Reordering generated slots before masked ones enables full KV cache reuse, while position IDs correspond to their ground-truth indices, invariant to reordering. Right (Training): Mirrors inference, optimizing a hybrid objective of autoregressive loss ($\mathcal{L}_{\text{ARM}}$) on permuted clean slots and denoising loss ($\mathcal{L}_{\text{MDM}}$) on masked slots.

where the term $\boldsymbol{R}_{n-m}$, which governs relative distance perception, remains invariant to token reorder. The aforementioned mechanism allows REFUSION to achieve full KV cache reuse, significantly boosting generation efficiency while maintaining flexibility in the global decoding order.

**Slot Partition for Reducing Learning Complexity.** We further elevate modeling granularity by partitioning the token sequence into continuous, non-overlapping *slots*. The design of both inter- and intra-slot decoding orders is a key consideration. Prior studies (Luxembourg et al., 2025) suggest that conditional independence assumptions break down most severely among nearby tokens. Our pilot dependency analysis (Appendix A.2) further confirms that dependency strength decays rapidly with relative distance, particularly in denser contexts with lower masking ratios. This motivates a hybrid decoding design: slots are generated in a diffusion-like manner with global flexibility, while tokens within them are decoded autoregressively to capture strong local dependencies.

This design naturally supports full KV cache reuse. For inter-slot decoding, after each decoding step, the newly decoded slots are moved in front of the remaining masked slots while preserving their internal order, corresponding to applying the previously described reordering process at the slot level. Within each slot, standard autoregressive KV caching is directly utilized during token-level decoding.

We intentionally use the term *slot* to distinguish our design from the well-known *block* partitioning in the literature. In contrast to our slot dynamics, block-based methods operate with inverted logic: they employ inter-block left-to-right autoregressive decoding and intra-block parallel diffusion decoding. For a detailed discussion on other fundamental distinctions, please refer to the paragraph "Comparison with Block-based Diffusion" in §2.

## 4.2 INFERENCE

Built upon the proposed sequence reorganization strategies, the inference process iteratively generates a final response $\tilde{r}_0$ given a prompt $p_0$, starting from an initial sequence $\tilde{r}_1$ composed of masked tokens. During inference, the response is partitioned into $K$ consecutive slots of length $k$. Each decoding iteration comprises two stages: diffusion-based slot selection and autoregressive slot infilling.

**Stage I: Diffusion-based Slot Selection.** This stage selects the next slots to be decoded in a diffusion-like manner. At timestep $t$, defined as the ratio of remaining masked slots, we construct the input sequence $\tilde{\boldsymbol{S}}_t$ by concatenating already-decoded slots ($\tilde{\boldsymbol{S}}_t^{\text{clean}}$, in their generation order) with the masked slots ($\tilde{\boldsymbol{S}}_t^{\text{masked}}$, in their original positional order).

The model then computes a certainty score for each masked slot based on its predictive distribution. While various scoring strategies exist, we adopt a simple yet effective one: the probability of the

most likely token at the slot's first position. An ablation of different scoring strategies is provided in Appendix C.2.

Finally, slots with scores exceeding a threshold $\tau_{\text{slot}}$ are selected for infilling. This strategy identifies slots that are highly predictable and suitable for parallel decoding given the current context. Furthermore, inspired by speculative decoding (Leviathan et al., 2023), we sample a corresponding draft $\tilde{S}_t^{\text{draft}}$ for these selected slots from the predictive distribution to accelerate the subsequent infilling stage.

**Stage II: Autoregressive Slot Infilling.**   The second stage begins by efficiently verifying the draft slots $\tilde{S}_t^{\text{draft}}$. Specifically, all draft slots are concatenated into a sequence according to their original positional order. The model performs a single forward pass to compute token probabilities conditioned on the prompt and the already-decoded slots. We then identify the longest continuous prefix of this concatenated sequence where every token's probability exceeds a threshold $\tau_{\text{token}}$. If this prefix covers one or more complete slots, these slots are accepted in their entirety. The remaining unverified or partially verified draft slots are re-masked, and the process immediately proceeds to the next selection iteration.

Otherwise, we revert to a parallel iterative completion process to handle the draft slots independently, following the spirit of speculative decoding again. Each iteration consists of two steps: *(i) Verification:* we independently identify the longest valid prefix for each slot where conditional probabilities exceed $\tau_{\text{token}}$; and *(ii) Prediction:* we retain these valid prefixes, re-mask the remaining suffixes, and concurrently predict the masked tokens using the model's MDM capability, conditioned on the context and accepted prefixes. This cycle repeats until all selected slots are fully decoded.

Finally, the newly completed slots are moved to the front of the remaining masked slots, following the reorganization in §4.1. Their KV caches are directly concatenated for future iterations. This serves as a valid approximation, as the lack of inter-slot conditioning during parallel generation has minimal impact on performance (see §C.3).

The select-and-infill cycle continues until all slots are completed, after which the response is constructed by restoring the original slot order. This process is formalized in Appendix A.3 and illustrated in Figure 3 (left). For further clarity, a step-by-step visualization is provided in Appendix D.2.

## 4.3   TRAINING

The training procedure for REFUSION is carefully designed to mirror the dynamics of our two-stage decoding algorithm. This requires a data construction strategy that simulates the non-sequential, partially-decoded states encountered during generation, and a hybrid training objective that jointly optimizes the model's selecting and infilling capabilities.

**Training Data Construction.**   To simulate the partially decoded states encountered during iterative generation, we introduce a three-step strategy to construct training data from each prompt-response pair $(p_0, r_0)$. The response $r_0$ is first partitioned into a sequence of $K$ slots, $\boldsymbol{S}_0 = [S_0^1, \ldots, S_0^K]$, each of size $k$. Then, a corrupted version $\boldsymbol{S}_t$ is constructed given a masking ratio $t \sim U(0,1)$ as follows: **(1) Random slot masking.** Analogous to token-level masking in traditional MDMs, we randomly select and mask $\lfloor tK \rfloor$ slots from the original sequence $\boldsymbol{S}_0$. Each selected slot is replaced with a block of $k$ "[MASK]" tokens. **(2) Permutation of clean slots.** Since the generation order of slots is dynamically determined, the model must learn to process context in any arbitrary permutation. To achieve this, we randomly permute the unmasked (clean) slots to form $\boldsymbol{S}_t^{\text{clean}}$, while keeping the original relative positions of the masked slots to form $\boldsymbol{S}_t^{\text{masked}}$. **(3) Reorder.** The final training instance $\boldsymbol{S}_t$ is assembled by concatenating the permuted clean slots followed by the masked slots.

**Hybrid Training Objective.**   To empower our model with the dual capabilities of global selecting and local infilling, we propose a hybrid training objective that learns from every token in the sequence.

On one hand, the clean slots $\boldsymbol{S}_t^{\text{clean}}$ are trained with a standard ARM loss for next token prediction:

$$\mathcal{L}_{\text{ARM}} = -\mathbb{E}_{\substack{(p_0, r_0) \sim \mathcal{D} \\ t \sim U(0,1)}} \left[ \frac{1}{(k-1) \cdot |\boldsymbol{S}_t^{\text{clean}}|} \sum_{i=1}^{|\boldsymbol{S}_t^{\text{clean}}|} \sum_{j=2}^{k} \log P_\theta \left( v_t^{i,j} \mid p_0, \boldsymbol{S}_{t,<(i,j)}^{\text{clean}} \right) \right], \qquad (2)$$

Table 1: Zero-shot performance and throughput (TPS) comparison on multiple benchmarks. Each model displays accuracy/pass@1 (top row) and throughput (TPS, bottom row). Within the MDM category, we highlight the best performance results in **bold** and underline the second best. An *italic* score in the ARM category signifies that it surpasses the best-performing MDM.

| Model | MMLU-Pro | ARC-C | GSM8K | MATH | GPQA | HumanEval | MBPP | Avg. |
|---|---|---|---|---|---|---|---|---|
| *Autoregressive Model* | | | | | | | | |
| **Llama-3-8B-Instruct** | 35.23 | 82.76 | 75.13 | 25.48 | 29.46 | 46.34 | 53.00 | 49.63 |
| | 32.07 | 44.12 | 42.81 | 19.73 | 42.00 | 42.26 | 41.68 | 37.81 |
| **Qwen3-8B** | *67.25* | *90.36* | *81.96* | *83.28* | *39.06* | *87.80* | 63.80 | *73.36* |
| | 31.42 | 42.78 | 31.20 | 30.11 | 30.43 | 30.95 | 30.07 | 32.42 |
| *Masked Diffusion Model* | | | | | | | | |
| **LLaDA-8B-Instruct** | 35.80 | 85.58 | 76.35 | 38.78 | 32.37 | 45.12 | 25.60 | 48.51 |
| | 18.21 | 0.03 | 27.35 | 23.93 | 1.99 | 12.42 | 2.97 | 12.41 |
| **LLaDA w/ Fast-dLLM** | 35.02 | 82.85 | 76.27 | 38.58 | 28.35 | 37.80 | 24.80 | 46.24 |
| | 39.81 | 0.86 | 73.07 | 52.23 | 17.54 | 62.52 | 37.19 | 40.46 |
| **LLaDA w/ D2F** | 22.84 | 84.13 | 39.04 | 23.68 | 31.25 | 36.59 | 35.20 | 38.96 |
| | 44.54 | 3.70 | 82.59 | 59.48 | 23.84 | 96.90 | 53.85 | 52.13 |
| **Dream-7B-Instruct** | 40.05 | 88.31 | 76.42 | 46.60 | 30.36 | 56.71 | 50.40 | 55.55 |
| | 15.98 | 0.06 | 20.30 | 18.99 | 1.81 | 3.51 | 1.23 | 8.84 |
| **Dream w/ Fast-dLLM** | 40.36 | 86.86 | 75.82 | 36.76 | 31.25 | 56.10 | 10.60 | 48.25 |
| | 47.18 | 1.42 | 61.49 | 58.24 | 22.96 | 49.73 | 19.55 | 37.22 |
| **Dream w/ D2F** | 23.82 | 85.92 | 41.62 | 29.98 | 30.13 | 50.00 | 51.60 | 44.72 |
| | 56.43 | 59.08 | 79.20 | 84.14 | 49.96 | 69.15 | 65.59 | 66.22 |
| **REFUSION** | **45.94** | **89.76** | **84.91** | **54.22** | **35.49** | **78.66** | **68.20** | **65.31** |
| | 52.74 | 32.46 | 81.24 | 81.77 | 64.11 | 103.90 | 92.09 | 72.62 |

where $v_t^{i,j}$ is the $j$-th token in the $i$-th clean slot, $S_{t,<(i,j)}^{\text{clean}}$ is the prefix of the token in $S_t^{\text{clean}}$.

On the other hand, the masked slots $S_t^{\text{masked}}$ are trained with an MDM objective for denoising[1]:

$$\mathcal{L}_{\text{MDM}} = -\mathbb{E}_{\substack{(p_0,r_0)\sim\mathcal{D} \\ t\sim U(0,1)}} \left[ \frac{1}{k \cdot |S_t^{\text{masked}}|} \sum_{i=1}^{|S_t^{\text{masked}}|} \sum_{j=1}^{k} \log P_\theta(v_0^{i,j} \mid p_0, S_t^{\text{clean}}, S_{t,\leqslant(i,j)}^{\text{masked}}) \right], \quad (3)$$

where $v_0^{i,j}$ is the ground-truth token from the original response corresponding to the $j$-th token in the $i$-th slot of $S_t^{\text{masked}}$. The final training objective is a summation of the two losses, balanced by $\lambda$:

$$\mathcal{L} = \mathcal{L}_{\text{ARM}} + \lambda \mathcal{L}_{\text{MDM}}. \quad (4)$$

Notably, this approach improves data efficiency, which contrasts with traditional MDMs where clean tokens only serve as context and provide no direct supervision.

Following prior work (Gong et al., 2025; Ye et al., 2025), we initialize $P_\theta$ with an off-the-shelf ARM backbone. Crucially, all tokens retain their original positional indices from $r_0$ throughout the training process. This allows the model to maintain awareness of the relative positions among all tokens, ensuring sequence coherence despite the shuffled input order. Figure 3 (right) illustrates the training process.

## 5 EXPERIMENTS

### 5.1 EXPERIMENTAL SETUP

**Implementation Details.** We initialize REFUSION from the Qwen3-8B checkpoint (Yang et al., 2025) and fine-tune it for 4 epochs on a diverse 3.7M-sample dataset ($\sim$1.22B tokens) covering mathematics, coding, and general instruction-following. For inference, leveraging the hierarchical compatibility with block-based diffusion methods discussed in §2, we define the block size as $b$. Implementation and hyperparameter details are provided in Appendix B.1 and B.2, respectively.

---

[1]Our per-token normalization, $\frac{1}{k \cdot |S_t^{\text{masked}}|}$, implicitly includes the $\frac{1}{t}$ weighting since $|S_t^{\text{masked}}| \approx tK$, where $K$ is the total number of slots.

**Evaluation Benchmarks and Metrics.** To comprehensively evaluate REFUSION, we test its performance on diverse benchmarks spanning: (1) General-purpose understanding and reasoning: MMLU-Pro (Wang et al., 2024) and ARC-C (Clark et al., 2018); (2) Mathematical and scientific problem-solving: GSM8K (Cobbe et al., 2021), MATH (Hendrycks et al., 2021), and GPQA (Rein et al., 2024); (3) Code generation: HumanEval (Chen et al., 2021) and MBPP (Austin et al., 2021). We use pass@1 for code generation and accuracy for the others. We further assess inference throughput in terms of tokens decoded per second (TPS) with a single A100 GPU and a batch size of 1.

**Baselines.** We evaluate REFUSION against three categories of baselines[2]: (1) ARMs: Llama-3-8B-Instruct (AI@Meta, 2024) and Qwen3-8B (Yang et al., 2025). (2) MDMs: LLaDA-8B-Instruct (Nie et al., 2025), and Dream-7B-Instruct (Ye et al., 2025). (3) State-of-the-art MDM acceleration methods: Fast-dLLM (Wu et al., 2025b) and D2F (Wang et al., 2025). We implement the baseline methods based on their official hyperparameters[3].

Table 2: Controlled comparison of models initialized from Qwen3-8B and trained on 120K subset.

| Model | MMLU-Pro | ARC-C | GSM8K | MATH | GPQA | HumanEval | MBPP | Avg. |
|---|---|---|---|---|---|---|---|---|
| **Qwen3-8B (Retrained)** | 54.22 | 90.53 | 88.17 | 66.94 | 30.36 | 63.72 | 66.00 | 65.71 |
| | 31.16 | 29.46 | 30.69 | 31.58 | 28.49 | 30.52 | 30.62 | 30.36 |
| **LLaDA (Retrained)** | 38.27 | 86.43 | 80.06 | 44.94 | 0.67 | 42.07 | 39.40 | 47.41 |
| | 1.89 | 0.02 | 9.73 | 13.92 | 0.21 | 2.57 | 1.31 | 4.24 |
| **BD3-LMs (Retrained)** | 26.11 | 78.24 | 83.55 | 47.32 | 29.02 | 59.15 | 46.80 | 52.88 |
| | 16.82 | 2.19 | 16.35 | 16.90 | 2.82 | 15.59 | 14.09 | 12.11 |
| **REFUSION (Retrained)** | 42.14 | 84.81 | 80.74 | 51.78 | 31.70 | 70.12 | 58.20 | 59.93 |
| | 40.56 | 29.01 | 53.96 | 77.42 | 46.65 | 58.40 | 67.35 | 53.34 |

Table 3: Controlled comparison with Dream-7B-Instruct on its native Qwen2.5-7B backbone.

| Model | MMLU-Pro | ARC-C | GSM8K | MATH | GPQA | HumanEval | MBPP | Avg. |
|---|---|---|---|---|---|---|---|---|
| **Dream-7B-Instruct** | 40.05 | 88.31 | 76.42 | 46.60 | 30.36 | 56.71 | 50.40 | 55.55 |
| | 15.98 | 0.06 | 20.30 | 18.99 | 1.81 | 3.51 | 1.23 | 8.84 |
| **REFUSION (Retrained)** | 35.25 | 83.11 | 80.21 | 46.36 | 29.02 | 68.90 | 61.60 | 57.78 |
| | 76.02 | 53.38 | 107.83 | 139.55 | 102.29 | 106.89 | 98.04 | 97.71 |

## 5.2 MAIN RESULTS

The main results in Table 1 highlight two key findings: **(1) REFUSION dominates all MDM baselines.** REFUSION consistently outperforms all MDM baselines in both performance and throughput (TPS) across all seven benchmarks, often by a substantial margin. For instance, on HumanEval, it achieves 78.66% pass@1, surpassing the next-best MDM (Dream-7B-Instruct) by nearly 22 absolute points. While acceleration methods like Fast-dLLM and D2F improve throughput at a significant performance cost, they still fall short of REFUSION's efficiency. Notably, on MBPP, REFUSION reaches 92.09 TPS, which is $1.4\times$ faster than the next-fastest MDM (Dream w/ D2F). REFUSION thus delivers both state-of-the-art performance and superior efficiency, establishing a new frontier for MDMs[4]. **(2) REFUSION challenges strong ARMs.** More remarkably, REFUSION challenges and often surpasses strong ARMs. It delivers an average speedup of $2.33\times$ over Qwen3-8B across all tasks while exhibiting superior performance on several benchmarks. For instance, on GSM8K and MBPP, it outperforms Qwen3-8B by 3.68 absolute points. This demonstrates that our non-autoregressive approach can break the long-standing trade-off between the speed of MDMs and the quality of ARMs (Feng et al., 2025).

## 5.3 CONTROLLED COMPARISON

To isolate the benefits of REFUSION from data or backbone advantages, we adopt a controlled approach: retraining reproducible baselines using unified settings, while retraining REFUSION to match non-open-source baselines.

---

[2]We omit BD3-LMs due to its limited scale (0.2B), but compare a scaled-up version (8B) in §5.3.

[3]Appendix C.1 compares a broader set of models.

[4]Appendix C.4 analyzes model trade-off frontiers.

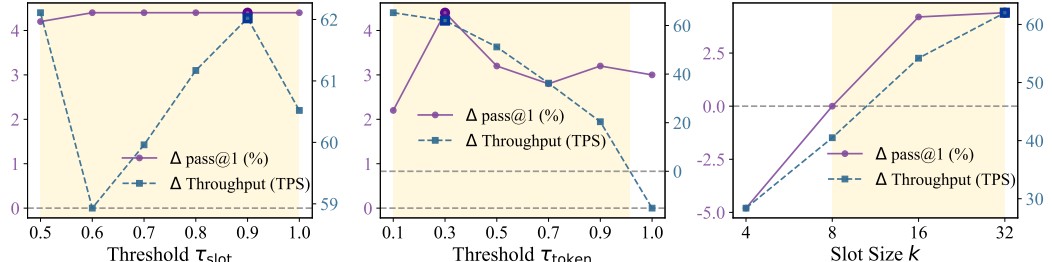

Figure 4: Impact of key hyperparameters on MBPP (0-shot). The plots show the change in **pass@1 (%)** and **throughput (TPS)** of REFUSION relative to Qwen3-8B (dashed lines at zero). When one parameter is varied, others are held at their default values ($\tau_{slot} = 0.9, \tau_{token} = 0.3, k = 32$). Yellow regions highlight the "sweet spot" where REFUSION surpasses the baseline in both metrics.

**Reproducible Baselines.** We conduct a controlled comparison using a 120K data subset randomly sampled from the full 3.7M dataset due to resource constraints. Specifically, we fine-tune Qwen3-8B, LLaDA, BD3-LMs[5], and REFUSION for 10 epochs using their respective original objectives, with all models initialized from Qwen3-8B. This setup ensures that observed differences are attributable solely to the algorithm design. Appendix C.5 discusses the scaling properties of REFUSION regarding data size.

Results in Table 2 confirm the architectural superiority of REFUSION. LLaDA suffers a catastrophic performance collapse, and BD3-LMs lags behind REFUSION in both performance and speed. We note that the already highly-optimized Qwen3-8B baseline understandably degrades when retrained on our smaller, open-source dataset. However, under this controlled setting (with data advantages eliminated), REFUSION still outperforms it by ∼6 points on HumanEval while being 1.9× faster. This result robustly validates that REFUSION's architectural innovations are the primary driver of its success, enabling effective learning even from limited data where standard MDMs fail.

**Non-Open-Sourced Baseline.** Furthermore, we conduct a controlled comparison with Dream-7B-Instruct. Since its training code and data processing details are not open-sourced, we cannot retrain it on Qwen3-8B. Instead, we train a REFUSION variant on Dream's original Qwen2.5-7B backbone. It is crucial to note the significant disparity in training resources: Dream benefits from massive pre-training (580B tokens, 146.5M samples) followed by SFT (1.8M samples), whereas our REFUSION variant is exclusively fine-tuned (3.7M samples) without any pre-training. Despite this disadvantage, Table 3 shows that REFUSION still achieves a 2.23% average performance gain and a massive 11.05× speedup over Dream. REFUSION significantly excels on reasoning and coding tasks (GSM8K, HumanEval, MBPP). Its lower performance on knowledge-intensive tasks (MMLU-Pro, ARC-C) is expected, as it skips the pre-training stage that Dream utilizes for knowledge injection. These results confirm that REFUSION's architectural advantages are robust across different base models and training setups.

### 5.4 ANALYSIS OF HYPERPARAMETERS

We examine the key hyperparameters governing the performance-efficiency trade-off in REFUSION: the slot selection threshold $\tau_{slot}$, the token acceptance threshold $\tau_{token}$, and the slot size $k$. The threshold $\tau_{slot}$ controls the confidence for slot selection, and $\tau_{token}$ governs the confidence for draft acceptance (infilling), while $k$ defines the granularity of the generation unit. An analysis of other hyperparameters is shown in Appendix C.7.

As illustrated in Figure 4, these hyperparameters create a predictable trade-off. **(1) Slot selection threshold $\tau_{slot}$:** Increasing $\tau_{slot}$ improves performance due to higher token reliability. However, throughput (TPS) exhibits a non-monotonic trend. Although a higher threshold reduces slot parallelism, processing fewer slots mitigates synchronization overhead during the parallel iterative completion phase, which can potentially boost throughput. **(2) Token acceptance threshold $\tau_{token}$:** Increasing $\tau_{token}$ reduces the number of accepted draft tokens per step, thereby lowering throughput.

---

[5]The block size is set to 8, which is the minimum block size used in the REFUSION experiments.

Performance also follows a non-monotonic trend. While higher thresholds enforce stricter verification, excessive strictness causes frequent truncation and regeneration. This can trap the model in local optima and degrade final performance. **(3) Slot size $k$:** A larger slot size $k$ enhances local coherence and allows more draft tokens to be accepted wholesale during verification. This leads to simultaneous gains in both performance and speed. Collectively, these analyses reveal a robust and wide "sweet spot," highlighted by the yellow shaded regions in Figure 4, where REFUSION consistently surpasses the Qwen3-8B baseline in both performance and throughput (TPS). This superior operating zone corresponds to a slot selection threshold $\tau_{\mathrm{slot}} \in [0.5, 1.0]$, a token acceptance threshold $\tau_{\mathrm{token}} \in [0.1, 0.9]$, and a slot size $k \in \{8, 32\}$.

### 5.5 CASE STUDY

Figure 5 provides a qualitative understanding of how REFUSION solves a programming problem from the MBPP benchmark, highlighting two key capabilities: **(1) High degree of parallelism.** The model frequently generates multiple slots concurrently. For instance, at iteration 8, it simultaneously generates four separate slots, demonstrating its ability to exploit parallel decoding opportunities. Simultaneously, within each slot, the model leverages speculative decoding to accept four tokens in a single pass, significantly accelerating the generation process. **(2) Non-linear generation order.** The generation process is markedly non-linear. For example, the model constructs the central "for" loop structure (iteration 5) before initializing a local variable "sum = 1" (iteration 6). This ability to select and execute in a parallel, non-monotonic fashion allows REFUSION to construct complex, structured code in a manner that is both efficient and conceptually closer to human problem-solving. Appendix D.1 shows the results of baseline models on the same problem.

Figure 5: A case study of REFUSION generating a Python function for an MBPP problem ($k = 4, \tau_{\mathrm{slot}} = 0.6, \tau_{\mathrm{token}} = 0.3, b = 16$). The number in the top-left corner of each slot indicates the generation order, while the color intensity within each slot represents the generation time (darker indicates earlier).

## 6 CONCLUSION

In this work, we present REFUSION, a novel generative model that leverages sequence reorganization with a causal attention framework. By synergizing the strengths of diffusion-based selection and autoregressive infilling, it effectively addresses the long-standing efficiency and coherence challenges in traditional MDMs. This unique design enables full KV cache reuse alongside flexible generation orders, while making the training objective tractable by simplifying the combinatorial complexity of the generation space. Extensive evaluations across seven benchmarks show that REFUSION establishes a new state of the art for MDMs. More strikingly, it bridges the performance gap to strong ARMs, often outperforming them while being significantly faster. Our work demonstrates that by structuring the parallel generation process, it is possible to achieve the throughput potential of MDMs without sacrificing generation quality. Future directions include further scaling of the model and data size, as well as leveraging reinforcement learning to optimize the model's planning policy for complex, multi-step reasoning tasks.

## ACKNOWLEDGMENTS

This work was supported by the Beijing Major Science and Technology Project under Contract no. Z251100008425002; the National Natural Science Foundation of China (Nos. 62522609, 92470118); the Beijing Natural Science Foundation (No. L247030); Ant Group Research Intern Program; and the fund for building world-class universities (disciplines) of Renmin University of China.

## REPRODUCIBILITY STATEMENT

To ensure the reproducibility of our experimental results, we have open-sourced our training and inference code. The specific settings for training and testing are detailed in §5.1 and Appendix B.

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

Table 4: Comparison between REFUSION and existing MDMs. $L$ denotes the generation length and $k$ denotes the block or slot size.

| Model | Generation Scope | Attention Mechanism | Generation Order | Full KV Cache Reuse | Number of Distinct Masking Patterns |
|---|---|---|---|---|---|
| **LLaDA** | Full Sequence | Bidirectional | Any-Order | ✗ | $\sum_{l=1}^{L} \binom{L}{l} \approx 2^L$ |
| **BD3-LMs** | Intra-block | Bidirectional | Any-Order | ✗ | $2^k \cdot \frac{L}{k}$ |
| | Inter-block | Causal | Left-to-Right | | |
| **REFUSION** | Intra-slot | Causal | Left-to-Right | ✓ | $\sum_{i=1}^{L/k} \binom{L/k}{i} \cdot i! = \lfloor (\frac{L}{k})! \cdot e \rfloor - 1$ |
| | Inter-slot | Causal | Any-Order | | $(\lfloor (\frac{L}{k})! \cdot e \rfloor - 1 \ll 2^L$ for large $k)$ |

# A   METHODOLOGICAL DETAILS

## A.1   COMPARISON BETWEEN REFUSION AND REPRESENTATIVE MDMS

Table 4 provides a detailed, side-by-side comparison of the architectural and methodological designs of REFUSION against two representative MDMs, LLaDA (Nie et al., 2025) and BD3-LMs (Arriola et al., 2025). This comparison highlights how REFUSION uniquely addresses the fundamental trade-offs between generation flexibility, computational efficiency, and learning complexity.

(1) LLaDA, as a conventional MDM, operates on the entire sequence with a bidirectional attention mechanism. This grants it maximum flexibility, allowing for a fully unconstrained, any-order generation process. However, this design choice incurs two significant penalties. First, the bidirectional attention is fundamentally incompatible with KV caching, resulting in substantial computational overhead at each decoding step. Second, it must learn dependencies across an exponential space of possible masking patterns. For a sequence of length $L$, any given training or inference state is defined by a subset of tokens that remain masked. Since each of the $L$ positions can be either masked or unmasked, the model must, in principle, handle any of the $2^L$ possible subsets of visible context[6]. This combinatorial space of approximately $2^L$ distinct masking patterns presents an intractable objective, as the model may not be sufficiently trained on the specific patterns encountered during inference, leading to incoherent parallel generation.

(2) BD3-LMs attempts to mitigate these issues with a hybrid, block-based approach. It enforces a rigid, left-to-right generation order between blocks, which enables KV cache reuse across block boundaries. However, within each block, it retains bidirectional attention and any-order token generation. This design makes a critical compromise. It sacrifices global generation flexibility, hindering the discovery of optimal generation strategies, which is a key theoretical advantage of MDMs. Furthermore, it still faces the challenges of token-level incoherence and the inability to utilize KV caching for intra-block decoding.

(3) REFUSION introduces a more elegant and unified solution. Generation is structured at the slot level. Within each slot (intra-slot), generation is autoregressive (left-to-right) under a causal attention mask, directly addressing the strong local dependencies between adjacent tokens. Between slots (inter-slot), the model retains the flexibility of any-order generation, enabling it to discover better, non-linear generation paths than the left-to-right order. Crucially, by reordering generated slots to always precede masked ones in the input sequence, REFUSION enables full KV cache reuse for every decoded token, a feature unique among these models. This design simultaneously achieves two critical goals: it combines global generation flexibility with universal computational efficiency, and it drastically reduces the learning complexity from an exponential token-level permutation space to a far more manageable slot-level one ($\lfloor (\frac{L}{k})! \cdot e \rfloor - 1$). For a typical sequence length of $L = 4,096$, a slot size of just $k = 8$ is sufficient to ensure $\lfloor (\frac{L}{k})! \cdot e \rfloor - 1 < 2^L$.

---

[6]Notably, due to the bidirectional attention, the model is invariant to the order in which clean tokens are revealed. Therefore, the learning complexity is not permutations ($L!$).

In summary, while prior models are forced to trade flexibility for efficiency or vice versa, REFUSION's innovative slot-based framework is the only approach that concurrently offers global any-order generation, full KV cache reuse, and a tractable training objective.

## A.2 LOCALITY OF INTER-TOKEN DEPENDENCY

A cornerstone of REFUSION is grouping contiguous tokens into slots to enable inter-slot parallelism and intra-slot serialization. This design is motivated by the critical insight that the conditional independence assumption is most prone to failure for nearby tokens, frequently leading to semantic incoherence (Luxembourg et al., 2025). To formalize this insight and guide our design, we conduct a pilot study to quantitatively investigate *how dependency strength between two tokens correlates with their relative distance*.

Formally, we define the dependency strength between two tokens, $x_0^i$ and $x_0^j$, in a given context $x_t$, as the degree to which the presence of $x_0^j$ influences the model's prediction of $x_0^i$. In practice, we approximate this measurement in a pilot study on the GSM8K test set (Cobbe et al., 2021). For a corrupted sequence $x_t$, we first reveal the ground-truth token $x_0^j$ at a randomly selected masked position $j$, and then quantify the influence of this reveal on the prediction at any other masked position $i$ through the Jensen-Shannon (JS) divergence (Manning & Schutze, 1999) between the distributions before and after this reveal, i.e., $p(x_0^i \mid x_t)$ and $p(x_0^i \mid x_t, x_0^j)$. A higher divergence implies stronger dependency, with zero divergence indicating conditional independence. Using both LLaDA (Nie et al., 2025) and Dream (Ye et al., 2025), we plot the averaged JS divergence against the relative distance between positions $i$ and $j$ in Figure 6. The average JS divergence consistently decays as the relative distance increases, and this decay is more rapid in denser contexts (i.e., lower masking ratios $t$).

## A.3 INFERENCE FORMALIZATION

In this section, we formalize the two-stage decoding iteration as follows. Crucially, standard causal attention is consistently applied throughout the decoding process, and the position IDs for all tokens always correspond to their indices in the original correct sequence, remaining invariant regardless of this reordering. By using these consistent position IDs with RoPE (Su et al., 2021), the model correctly perceives relative positions even when the input buffer is reordered.

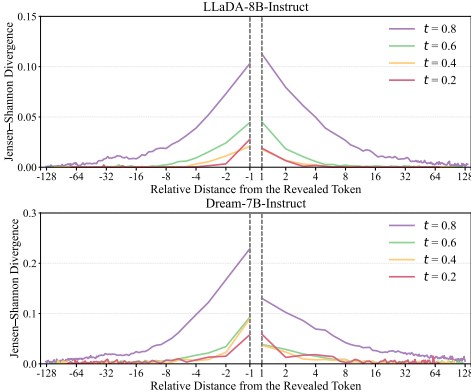

Figure 6: The locality of inter-token dependency in MDMs, with the sign on the x-axis denoting the direction from the revealed token (positive for rightward, negative for leftward).

**Stage I: Diffusion-based Slot Selection.** The first stage leverages the model's MDM capability to select the next decoding slots. At a timestep $t$ (defined as the ratio of remaining masked slots), we construct the input $\tilde{S}_t$ for enabling KV cache by concatenating already-decoded clean slots ($\tilde{S}_t^{\text{clean}}$, in generation order) with the remaining masked slots ($\tilde{S}_t^{\text{masked}}$, in their original positional order). The selection process then generates a draft for all masked slots. This draft serves a dual purpose: providing a basis for scoring each slot for selection, and acting as a speculative guess for the subsequent infilling stage (Leviathan et al., 2023). Specifically, for each position $j$ in the $i$-th slot of $\tilde{S}_t^{\text{masked}}$, a draft token $\tilde{d}_t^{i,j}$ is sampled from the model's marginal distribution, conditioned on the leading context:

$$\tilde{d}_t^{i,j} \sim P_\theta(\cdot \mid p_0, \tilde{S}_t^{\text{clean}}, \tilde{S}_{t,\leqslant(i,j)}^{\text{masked}}), \tag{5}$$

where $\tilde{S}_{t,\leqslant(i,j)}^{\text{masked}}$ denote the tokens before the position of the target token. This yields a draft version of the masked slots, denoted as $\tilde{S}_t^{\text{draft}} = \{\tilde{d}_t^{i,j}\}$. We then quantify the model's certainty score of $i$-th slot $\tilde{S}_t^i$ in $\tilde{S}_t^{\text{masked}}$ as the model's predicted probability of its first token $\tilde{d}_t^{i,1}$:

$$\mathcal{C}(\tilde{S}_t^i) = P_\theta(\tilde{d}_t^{i,1} \mid p_0, \tilde{S}_t^{\text{clean}}, \tilde{S}_{t,\leqslant(i,1)}^{\text{masked}}). \tag{6}$$

The model then selects a batch of slots with scores exceeding a threshold $\tau_{\text{slot}}$ for subsequent infilling. If no slot meets this criterion, the single slot with the globally highest score is selected instead. This strategy identifies slots that are strongly constrained by the existing context and weakly interdependent (e.g., distinct function definitions in code generation), making them suitable to parallelize.

**Stage II: Autoregressive Slot Infilling.** The second stage verifies and completes the selected draft slots by leveraging the model's autoregressive modeling capability. To accelerate infilling, we employ a speculative decoding strategy powered by the model's MDM capability. To achieve this, we first concatenate the slots in their original left-to-right order. The model then calculates the conditional probability of each token, conditioned on all preceding tokens within the newly formed sequence:

$$
\mathcal{P}(\tilde{d}_t^{i,j}) = \begin{cases} P_\theta(\tilde{d}_t^{i,1} \mid p_0, \tilde{\boldsymbol{S}}_t^{\text{clean}}, \tilde{\boldsymbol{S}}_{t,\leqslant(i,1)}^{\text{masked}}), & \text{if } j = 1 \\ P_\theta(\tilde{d}_t^{i,j} \mid p_0, \tilde{\boldsymbol{S}}_t^{\text{clean}}, \tilde{\boldsymbol{S}}_{t,<(i,j)}^{\text{draft}}). & \text{if } j > 1 \end{cases} \tag{7}
$$

Next, we verify the draft by identifying the longest prefix of the concatenated sequence, denoted as length $l$, where every token's probability exceeds the threshold $\tau_{\text{token}}$. If this prefix spans at least one complete slot (i.e., $l \geqslant k$), we accept the first $\lfloor l/k \rfloor$ slots in their entirety. The remaining unverified or partially verified slots are then re-masked, allowing the model to immediately initiate a new select-and-infill iteration while bypassing the costly suffix completion.

Otherwise, if the accepted prefix fails to cover at least one complete slot (i.e., $l < k$), we revert to processing the draft slots independently via a **Parallel Iterative Completion** process. We independently refine the draft $\tilde{S}_t^i = \{\tilde{d}_t^{i,j}\}_{j=1}^k$ for each slot $i \in \mathcal{I}$ in parallel, where $\mathcal{I}$ denotes the set of indices for the selected slots. The completion process iterates through two steps until all selected slots are fully completed. In each iteration $m$:

*(i) Verification:* For each slot $i \in \mathcal{I}$, we identify the longest valid prefix length $l_i^{(m)}$ where the conditional probability of every token in the prefix exceeds $\tau_{\text{token}}$, conditioned on the prompt, the already-decoded clean slots, and the preceding tokens within slot $i$:

$$
l_i^{(m)} = \max \left\{ \eta \;\middle|\; \forall j \leq \eta, \; P_\theta(\tilde{d}_t^{i,j} \mid p_0, \tilde{\boldsymbol{S}}_t^{\text{clean}}, \tilde{d}_t^{i,<j}) > \tau_{\text{token}} \right\}, \tag{8}
$$

where $\tilde{d}_t^{i,<j}$ denotes the prefix within the $i$-th slot.

*(ii) Prediction:* We retain the valid prefix $\tilde{d}_t^{i,1:l_i^{(m)}}$, re-mask the remaining suffix, and predict the masked tokens by sampling from the model's distribution:

$$
\tilde{d}_t^{i,j} \sim P_\theta(\cdot \mid p_0, \tilde{\boldsymbol{S}}_t^{\text{clean}}, \tilde{d}_t^{i,1:l_i^{(m)}}, \tilde{\boldsymbol{S}}_{t, \, l_i^{(m)} < \cdot \leqslant j}^{\text{masked}}), \quad \forall j \in (l_i^{(m)}, k]. \tag{9}
$$

After infilling each selected slot, the completed slots are moved from $\tilde{\boldsymbol{S}}_t^{\text{masked}}$ to $\tilde{\boldsymbol{S}}_t^{\text{clean}}$. For the subsequent iteration, the KV caches from these parallel-generated slots are concatenated. While this parallel generation forgoes inter-slot conditioning, we observe in our experiments that this has a minimal impact on final performance (see §C.3). This select-and-infill iteration repeats with an updated timestep $t$ until no masks remain ($t = 0$), at which point the final response $\tilde{r}_0$ is formed by sorting $\tilde{\boldsymbol{S}}_0^{\text{clean}}$ back into its original sequence order.

# B EXPERIMENTAL DETAILS

## B.1 IMPLEMENTATION DETAILS

Our training data comprises 3.7M samples from MAmmoTH (Yue et al., 2023), OpenMathInstruct-2 (Toshniwal et al., 2024), OpenCoder (Huang et al., 2024), SmolLM 2 (Allal et al., 2025), and Tulu 3 (Lambert et al., 2024). For OpenMathInstruct-2, we use its 1M open-source version and remove questions longer than 1,024 tokens as instructed. We use a global batch size of 512, a maximum sequence length of 4,096, and a learning rate of 2e-5. The training is conducted on 16 nodes, each with 8 H20 GPUs, and is accelerated using DeepSpeed ZeRO-2 (Rajbhandari et al., 2020) and Flash-attention-2 (Dao, 2023). The total training cost is approximately 10.68K H20 GPU-hours. We set $\lambda$ in Eq. 4 to 1. For each training sample, we randomly select a slot size from $\{4, 8, 16, 32\}$.

Table 5: Hyperparameter settings for different tasks.

| Benchmark | Generation Length | Slot Selection Threshold $\tau_{slot}$ | Token Acceptance Threshold $\tau_{token}$ | Slot Size $k$ | Block Size $b$ |
|---|---|---|---|---|---|
| MMLU-Pro | 512 | 0.9 | 0.4 | 16 | 128 |
| ARC-C | 512 | 0.8 | 0.1 | 8 | 8 |
| GSM8K | 512 | 0.9 | 0.4 | 32 | 128 |
| MATH | 512 | 0.8 | 0.6 | 32 | 64 |
| GPQA | 512 | 0.8 | 0.2 | 8 | 16 |
| HumanEval | 512 | 0.9 | 0.4 | 32 | 128 |
| MBPP | 512 | 0.9 | 0.3 | 32 | 128 |

Table 6: Performance comparison of different models on HumanEval and MBPP.

| Model | # Total Params | # Activated Params | System Optimization | Throughput (TPS) | HumanEval (# Shots) | MBPP (# Shots) |
|---|---|---|---|---|---|---|
| *Autoregressive Model* | | | | | | |
| Nova Micro | - | - | ✓ | 148 | 79.30 (0) | 65.40 (3) |
| GPT 4o Mini | - | - | ✓ | 59 | 88.00 (0) | 74.60 (3) |
| Gemini 2.0 Flash Lite | - | - | ✓ | 201 | 90.00 (0) | 75.00 (3) |
| *Masked Diffusion Model* | | | | | | |
| Mercury Coder Mini | - | - | ✓ | 1,109 | 88.00 (0) | 77.10 (3) |
| Mercury Coder Small | - | - | ✓ | 737 | 90.00 (0) | 76.60 (3) |
| Gemini Diffusion | - | - | ✓ | 1,479 | 89.60 (0) | 76.00 (3) |
| Seed Diffusion | - | - | ✓ | 1,600 | 82.80 (0) | 79.40 (3) |
| LLaDA-MoE | 7B | 1B | ✓ | 884 | 62.20 (0) | 67.45 (3) |
| REFUSION | 8B | 8B | ✗ | 98 | 78.66 (0) | 68.20 (0) |

Existing MDMs decode sequences to a predetermined length. Even when an end-of-sequence (EOS) token appears early, the model still expends decoding time on all tokens with higher position IDs. To address this issue, we introduce a mechanism for efficient variable-length generation. Specifically, during training, we pad shorter sequences in a mini-batch with padding tokens and exclude these tokens from the loss computation. During inference, upon generating an EOS token, we dynamically truncate the target length to that token's position. This prevents the decoding of any tokens with a higher position ID, thereby reducing redundant computation.

## B.2 Hyperparameter Setting

During the REFUSION inference process, four hyperparameters can be adjusted: the slot selection threshold $\tau_{slot}$, the token acceptance threshold $\tau_{token}$, the slot size $k$, and the block size $b$. Table 5 shows the specific settings used in our evaluation.

## C Experiment Results

### C.1 Comparison with Closed-Source and Diverse Architectures

To comprehensively demonstrate our advantages, we further compare REFUSION against closed-source and diverse ARM and MDM architectures. As shown in Table 6, REFUSION outperforms several highly optimized ARMs in either speed or performance. Specifically, it achieves $1.66\times$ the throughput of GPT-4o Mini and surpasses Nova Micro by 2.8 points on MBPP. Furthermore, REFUSION yields results approaching those of larger and more powerful closed-source MDMs.

### C.2 Ablation on Certainty Scoring

A key design choice in our slot selection stage is the metric used to compute the certainty score, which determines which slots are selected for parallel generation. In our default implementation, as described in Section A.3, we use the probability of the most likely token at the slot's first position. This choice is specifically motivated by our two-stage decoding process: the diffusion-based stage aims to identify valid anchors for parallel generation, while the subsequent autoregressive infilling stage ensures local coherence by completing the slot conditioned on this initial token. Therefore, the confidence of the first token serves as an efficient and effective proxy for the viability of initiating a slot's generation.

Table 7: Comparison of certainty scoring strategies on zero-shot performance. ↑ (↓) indicates that a higher (lower) value represents greater certainty. "Prob. of First Token" denotes our default method (used in Table 1), while "Mean Prob. of Slot" and "Mean Entropy of Slot" serve as alternatives. The highly comparable results across metrics validate our design choice.

| Method | MMLU-Pro | ARC-C | GSM8K | MATH | GPQA | HumanEval | MBPP |
|---|---|---|---|---|---|---|---|
| Prob. of First Token ↑ | 45.94 | 89.76 | 84.91 | 54.22 | 35.49 | 78.66 | 68.20 |
| Mean Prob. of Slot ↑ | 44.96 | 89.76 | 82.71 | 54.18 | 34.60 | 78.66 | 68.00 |
| Mean Entropy of Slot ↓ | 41.95 | 89.76 | 83.09 | 54.22 | 33.71 | 77.44 | 66.80 |

Table 8: Ablation regarding our KV cache reuse mechanism. We compare our default REFUSION, which efficiently reuses KV caches by concatenating them after parallel generation, against a variant (w/ KV Re-computation) that recomputes caches for full contextualization at a higher cost.

| Model | MMLU-Pro | ARC-C | GSM8K | MATH | GPQA | HumanEval | MBPP |
|---|---|---|---|---|---|---|---|
| REFUSION w/ KV Re-computation | 45.56 | 89.76 | 84.38 | 54.18 | 35.49 | 77.44 | 68.20 |
|  | 42.80 | 28.03 | 69.42 | 69.20 | 48.51 | 78.00 | 74.45 |
| REFUSION | 45.94 | 89.76 | 84.91 | 54.22 | 35.49 | 78.66 | 68.20 |
|  | 52.74 | 32.46 | 81.24 | 81.77 | 64.11 | 103.90 | 92.09 |

Alternatively, more intuitive approaches might involve using the mean probability or average entropy of the most likely tokens across all positions within a draft slot. To evaluate these alternatives, we conduct a comparative experiment. As shown in Table 7, the performance of all three methods is highly comparable across seven benchmarks. This consistency suggests that these metrics are robust and likely identify a significantly overlapping set of high-confidence slots. Given this performance parity and the lower computational overhead of the first-token probability, we retain it as our default method.

## C.3 ABLATION ON KV CACHE REUSE

To maximize efficiency, REFUSION directly concatenates the KV caches of parallel-generated slots, bypassing a costly forward pass that would otherwise be needed to contextualize them. To quantify the impact of this approximation, we conduct an ablation study comparing our default model against a variant, "REFUSION w/ KV Re-computation," which performs this extra forward pass to ensure full contextualization at the cost of speed.

As shown in Table 8, our default approach is consistently $1.16-1.33\times$ faster across all benchmarks. Surprisingly, this significant speedup comes at virtually no cost to performance; in fact, accuracy remains stable and even slightly improves on several benchmarks. We hypothesize this counter-intuitive benefit arises from a form of implicit regularization: by avoiding over-conditioning on potentially flawed parallel drafts, our method mitigates error propagation. This result validates our KV cache reuse strategy not merely as a speed-accuracy trade-off, but as a design choice that simultaneously enhances efficiency and robustness.

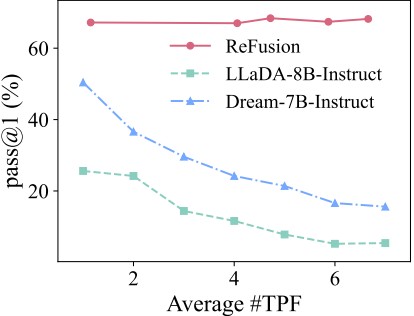

Figure 7: Pass@1 on MBPP for REFUSION and baseline MDMs over TPF.

## C.4 TRADE-OFF FRONTIER ANALYSIS

To further distinguish REFUSION from prior MDMs, we investigate the trade-off frontiers of different models. Figure 7 shows that both LLaDA and Dream suffer a sharp performance decline as parallelism (tokens generated per forward pass, TPF) increases[7], indicating a failure to uphold the conditional independence assumption when selecting tokens for parallel decoding. In contrast, REFUSION's

---

[7]We use TPF here, rather than TPS, to isolate the algorithmic trade-off from any system-level overheads.

curve is substantially flatter, validating that its training and decoding strategies can more reliably identify conditionally independent tokens.

## C.5 SCALING WITH DATA SIZE

To understand the scaling properties of our model, we investigate the impact of training data size on REFUSION's performance and efficiency. To this end, we collect additional data of potentially lower quality to expand our training set to 14M samples. Figure 8 illustrates the results of this analysis on GSM8K and MBPP, where we train REFUSION for one epoch on datasets of varying sizes (from 120K to 14M samples) and evaluate it using the same hyperparameters as in Table 5.

The results reveal a clear and positive scaling trend for both key metrics. Specifically, throughput (TPS, dashed lines) improves generally consistently as the training data size increases. For instance, on MBPP, throughput rises from approximately 51 TPS with 120K samples to over 81 TPS with 14M samples. This indicates that as the model is exposed to more diverse data, its internal generation process becomes more efficient, likely due to higher confidence and thus a higher acceptance rate of its parallel drafts, leading to fewer decoding iterations.

Interestingly, the performance scaling (solid lines) is not strictly monotonic, a common phenomenon when training with a fixed epoch count. On GSM8K, accuracy peaks at 2M samples before slightly decreasing at 3.7M. This behavior highlights a trade-off between data breadth and training depth under a constrained computational budget: with a fixed one-epoch schedule, training on a larger dataset potentially leads to under-convergence relative to the dataset's complexity.

Nevertheless, the consistent rise in throughput coupled with the substantial performance uplift from the 120K baseline suggests that with an increased computational budget (i.e., more training epochs on the larger datasets), performance would likely continue to improve, further unlocking the full potential of our approach.

## C.6 ANALYSIS OF GENERATION LENGTH AND LATENCY

To ensure that REFUSION's superior performance is not merely an artifact of generating more tokens than baselines, we present a direct comparison of the average generated token length and total inference latency for representative tasks. The results, shown in Table 9, address the hypothesis that quality gains might stem from quantity.

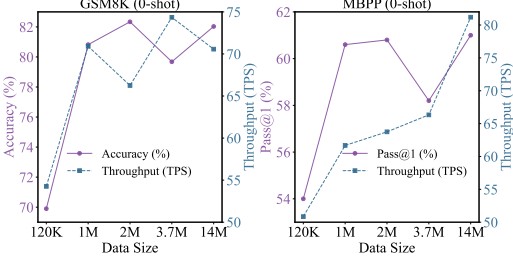

Figure 8: Scaling properties of REFUSION on GSM8K and MBPP. We plot performance (Accuracy/Pass@1, %) and inference throughput (TPS) as a function of training data size.

The data clearly demonstrates that REFUSION's generated outputs are consistently and significantly shorter than those of the ARM baseline (Qwen3-8B) and are either shorter or comparable in length to other MDMs. For instance, on MMLU-Pro, REFUSION generates only 124 tokens, roughly 5× fewer than Qwen3-8B, while achieving superior performance. This directly refutes the hypothesis that our model's quality gains are achieved by generating longer sequences.

Furthermore, the table highlights REFUSION's dramatic efficiency advantage, with measured latency being substantially lower across all tasks. These results confirm that REFUSION's superior performance-efficiency profile is a direct result of its methodological innovations, enabling it to produce concise and high-quality responses with minimal latency.

## C.7 ANALYSIS OF BLOCK SIZE

Our inference strategy is compatible with block-based diffusion methods (Arriola et al., 2025). Specifically, during inference, the target sequence is partitioned into consecutive blocks of size $b$. These blocks are decoded sequentially, with our decoding algorithm applied within each block. Notably, the constraint $b \geqslant k$ must be satisfied, where $k$ is the size of a slot, the fundamental unit for parallel decoding in our method.

Table 9: Comparison of average generated length and total latency (in seconds) across key benchmarks. Each model displays generated length (top row) and the measured latency (bottom row).

| Model | MMLU-Pro | ARC-C | GSM8K | MATH | GPQA | HumanEval | MBPP |
|---|---|---|---|---|---|---|---|
| **Qwen3-8B** | 654 | 62 | 300 | 778 | 810 | 444 | 53 |
| | 20.82s | 1.46s | 9.63s | 25.84s | 26.59s | 14.34s | 1.74s |
| **LLaDA-8B-Instruct** | 251 | 1 | 247 | 242 | 9 | 334 | 82 |
| | 13.81s | 32.86s | 9.02s | 10.13s | 4.71s | 26.85s | 27.65s |
| **Dream-7B-Instruct** | 211 | 2 | 223 | 206 | 10 | 121 | 45 |
| | 13.23s | 33.74s | 10.99s | 10.87s | 5.39s | 34.36s | 36.26s |
| **REFUSION** | 124 | 2 | 141 | 177 | 9 | 65 | 45 |
| | 2.35s | 0.07s | 1.74s | 2.17s | 0.14s | 0.63s | 0.48s |

Figure 9 illustrates the impact of block size $b$ on our method's performance and throughput (TPS). The figure reveals that both metrics exhibit non-monotonic trends as $b$ increases. The trend in performance arises from a trade-off between the global selection horizon and modeling complexity. Specifically, while a larger $b$ allows the model to cover longer semantic units in a single selection to enhance coherence, it simultaneously imposes a greater challenge in generating larger, more complex blocks in an arbitrary order. Similarly, the non-monotonic trend in throughput (TPS) is attributable to computational overhead. Although a larger $b$ provides more opportunities for parallelism, it forces the model to process a longer sequence containing many "padded" (i.e., yet-to-be-generated) positions. This significantly increases the latency of each decoding step, which eventually diminishes and then reverses the throughput gains observed with larger block sizes.

Although performance fluctuates slightly with larger block sizes, REFUSION's pass@1 decreases by only approximately 0.2% relative to its peak as $b$ increases from 32 to 512. This robustness to block size highlights the model's ability to leverage strong diffusion-based selection to select the most appropriate slots for decoding across a wide range. Collectively, these analyses reveal a robust and wide "sweet spot," highlighted by the yellow shaded regions in Figures 4 and 9, where REFUSION consistently surpasses the Qwen3-8B baseline in both performance and throughput (TPS). This superior operating zone corresponds to a block size $b \in [32, 128]$.

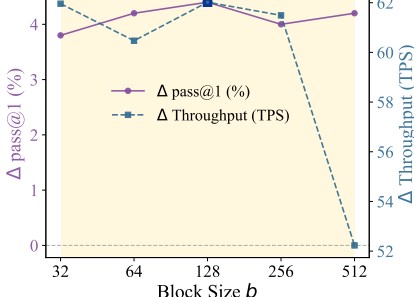

Figure 9: Relative change in REFUSION's **pass@1 (%)** and **throughput (tokens/sec)** compared to Qwen3-8B (horizontal dashed lines at zero) as block size $b$ varies. The yellow shaded region highlights the range of $b$ where REFUSION surpasses Qwen3-8B.

## D CASE STUDY

### D.1 CODE GENERATION CASE STUDY

Table 10 showcases responses to the same MBPP problem as Figure 5 from different models. REFUSION's ability to globally sketch the high-level structure via a diffusion-based process while locally infilling details autoregressively results in better-structured, high-quality code, demonstrating its superiority over existing MDMs.

### D.2 STEP-BY-STEP VISUALIZATION OF INFERENCE

In order to facilitate the understanding of our inference method, we show a step-by-step decoding process in Figure 10. Specifically, our inference process progressively generates the response through an iterative "select-and-infill" mechanism.

The model maintains standard token-wise causal attention throughout the entire process. Each decoding iteration operates as a two-stage cycle: First, the selection stage predicts drafts for all masked slots in parallel and selects a subset of high-quality slots based on confidence scores. Second, the infilling stage treats the selected slots as a batch and completes them autoregressively to ensure local coherence.

Table 10: Model responses to the same MBPP problem as Figure 5.

| Problem | You are an expert Python programmer. Your task is to write a single Python function to solve the problem described below, and here is your task: Write a function to sum all amicable numbers from 1 to a specified number. Your code should pass these tests: |
|---|---|

```
assert amicable_numbers_sum(999)==504
assert amicable_numbers_sum(9999)==31626
assert amicable_numbers_sum(99)==0
```

**LLaDA-8B-Instruct** ✗

```python
def amicable_numbers_sum(limit):
    def is_perfect_square(n):
        root = int(n**0.5)
        return root * root == n

    def sum_divisors(n):
        total = 0
        for i in range(1, n):
            if n % i == 0:
                total += i
        return total

    def is_amicable(a, b):
        return sum_divisors(a) == b and sum_divisors(b) == a

    amicable = set()
    for i in range(1, limit + 1):
        if is_perfect_square(i):
            amicable.add(i)
        for j in range(i + 1, limit + 1):
            if is_amicable(i, j):
                amicable.add(j)

    return sum(amicable)
```

**Dream-7B-Instruct** ✗

```python
def amicable_numbers_sum(n):
    def is_amicable(num):
        divisors = 1
        for i in range(2, num):
            if num % i == 0:
                divisors += i
                if i != num // i:
                    divisors += num // i
        return divisors == num

    total = 0
    for i in range(2, n):
        if is_amicable(i):
            total += i
    return total
```

**REFUSION** ✓

```python
def amicable_numbers_sum(n):
    def sum_divisors(num):
        sum = 1
        for i in range(2, int(num**0.5) + 1):
            if num % i == 0:
                sum += i
                if i != num // i:
                    sum += num // i
        return sum

    amicable_sum = 0
    for i in range(2, n + 1):
        sum_i = sum_divisors(i)
        if sum_i != i and sum_divisors(sum_i) == i:
            amicable_sum += i
    return amicable_sum
```

To enable full KV cache reuse, the newly decoded slots are moved to the front of the remaining masked slots after infilling. Crucially, while the tokens' positions within the input sequence may change due to this reordering, their position IDs remain invariant, always corresponding to their

indices in the original correct sequence. By utilizing these consistent position IDs with RoPE (Su et al., 2021), the model accurately perceives the relative positions of all tokens, even when the input buffer is reordered.

# E    LIMITATIONS

A primary limitation of our current framework is the immutability of generated slots. Once the tokens within a slot are generated via diffusion-based selection and autoregressive infilling, they are considered final and cannot be remasked or refined in future iterations. This design choice, while simplifying the process, precludes the model from correcting potential errors made within a completed slot.

A promising direction for future work would be to introduce a re-masking mechanism at the sub-slot level. For instance, after infilling a slot, the model could verify the generated tokens and preserve only a high-confidence prefix, while re-masking the lower-quality suffix. This would allow for iterative refinement but would necessitate a more complex inference logic, potentially involving dynamic adjustments of slot sizes to handle these newly masked, smaller segments. Developing an efficient strategy for such dynamic, fine-grained refinement remains a key challenge for future research.

# F    THE USE OF LARGE LANGUAGE MODELS

In the interest of complete transparency, we wish to clarify the use of AI assistance in the preparation of this manuscript. The core research ideas, including the conception of the REFUSION model, the design of the training and inference algorithms, all experimental setups, and the analysis of the results were developed exclusively by the human authors. We utilized a Large Language Model for the limited purpose of linguistic refinement. This involved polishing certain sentences and paragraphs to improve grammatical correctness, clarity, and overall flow. This usage was restricted to editing and did not extend to research ideation, content generation, or experimental analysis.

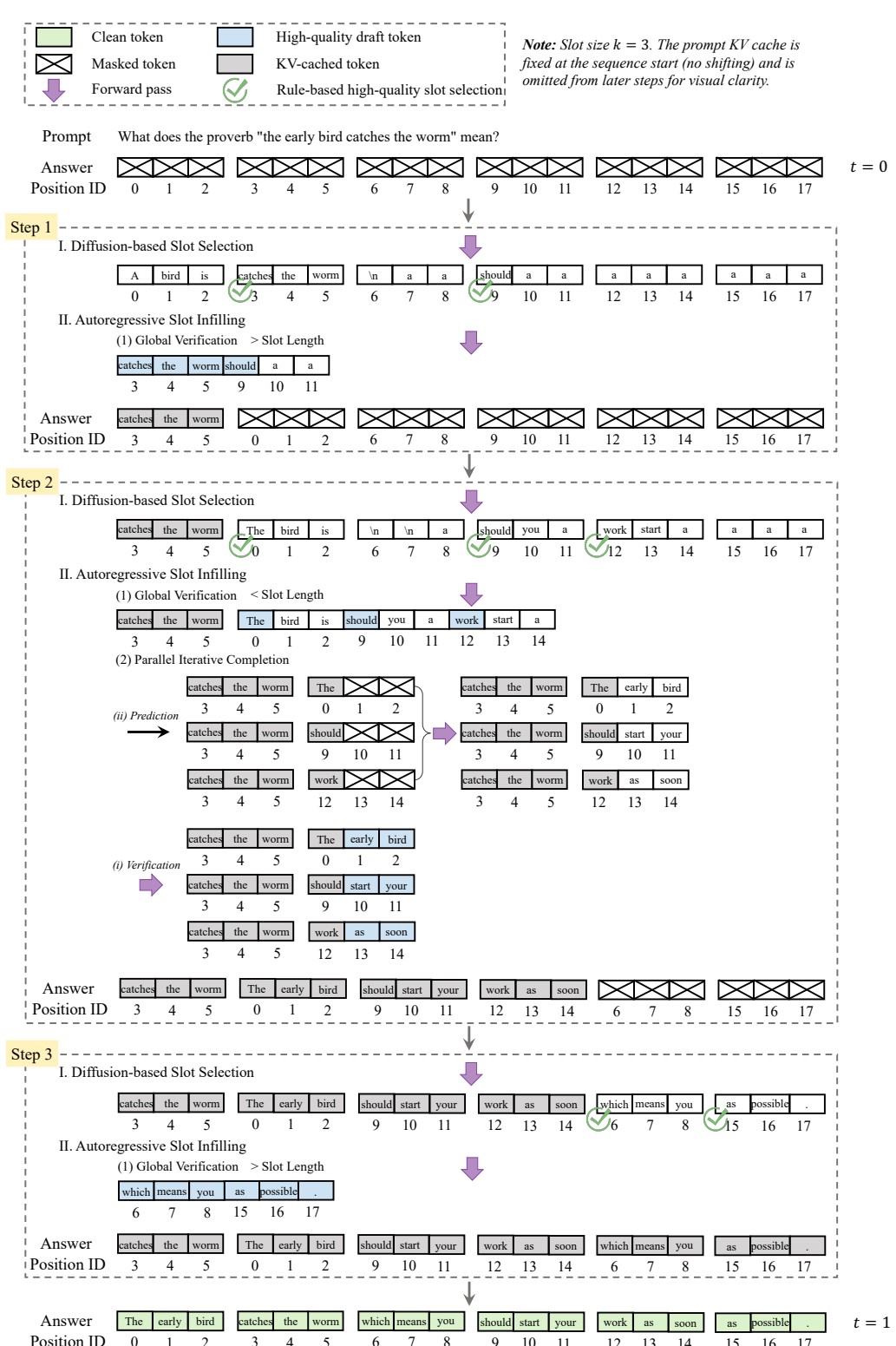

Figure 10: Visualization of the REFUSION inference mechanism.

