# OpenReview forum: "ReFusion: A Diffusion Large Language Model with Parallel Autoregressive Decoding"
_ICLR.cc/2026/Conference — ICLR 2026 Poster_

### Official Review · Reviewer_wAnP · 2025-10-28

**Soundness:** 2
**Presentation:** 1
**Contribution:** 3
**Rating:** 4
**Confidence:** 3

**Summary:**

The authors propose ReFusion, a hybrid generative model that integrates inter-slot parallel decoding with intra-slot autoregressive decoding. ReFusion is the first Masked Diffusion Model (MDM) to achieve full KV cache reuse for every decoded token, a significant efficiency gain, while maintaining global generation flexibility and a tractable training process. Based on their experiments, the authors assert that ReFusion dominates all MDM baselines and is competitive with autoregressive models (ARMs).

**Strengths:**

* The paper combines the strengths of diffusion and autoregressive generation, helping to maximize parallelism and quality.

* The idea of achieving full KV caching for every token within a parallel generation framework is significant and worthwhile.

* The paper provides an interesting analysis regarding the locality of inter-token dependency.

* The reported experimental results are strong, showing performance that Pareto-dominates existing baselines.

**Weaknesses:**

* The proposed method appears conceptually similar to Eso-LM [1]. At a high level, both models seem to perform diffusion and autoregressive infilling. The authors must do more to clearly and substantially differentiate ReFusion from this existing work.

* The paper lacks clarity regarding the training and architectural choices. It is not at all clear how the model achieves "full KV cache reuse of every decoded token" (lines 103-104). Diffusion LLMs typically require bidirectional attention to support flexible, any-order generation, which inherently prevents KV caching. However, Table 4 claims ReFusion supports any-order generation while using a causal mask. It is very difficult to understand how the model can simultaneously support parallel sampling and use a causal mask for KV caching. This mechanism would be made more clear with a detailed example of a training input and its corresponding attention mask.

* The illustrative example provided in Section 5.5 is very confusing and fails to clarify the generation process. It is not clear which text segments were generated via diffusion and which were generated autoregressively.

* The manuscript reads as heavily LLM influenced, but no disclosure is made.

[1] Sahoo, Subham Sekhar, et al. "Esoteric Language Models." arXiv preprint arXiv:2506.01928 (2025).

**Questions:**

1. In your discussion of Eso-LM, you state: “dynamically reposition newly generated tokens ahead of masked ones at each step to facilitate caching. However, this strategy introduces an intractable learning objective at a token-level permutation space” (lines 130-132). Can you please clarify what this "intractable learning objective" is and why it is intractable? What makes ReFusion different?

2. For the experimental benchmarks, how many tokens are being generated for each task? Is it possible that ReFusion achieves better quality simply by generating more tokens than the baselines?

3. What are the measured latencies for ReFusion and the baseline models?

---

> ### Author Response · Authors · 2025-11-21
> **Response to Reviewer wAnP (Part 1)**
>
> We thank the reviewer for the constructive feedback. For clarity, we address Weakness 2 first to establish our methodology, laying the groundwork for the comparisons in Weakness 1.
>
> > Wk2: The paper lacks clarity regarding the training and architectural choices. It is very difficult to understand how the model can simultaneously support parallel sampling and use a causal mask for KV caching.
>
> 1. The Solution to the Paradox: Full KV Cache Reuse with Any-Order Generation
>
> 	We thank the reviewer for the opportunity to clarify our method, which indeed resolves the paradox of using a causal mask for **KV cache reuse** while simultaneously **supporting any-order generation**.
>
> 	**Mechanism:** We maintain any-order flexibility by not constraining the model’s generation order. To support KV cache reuse simultaneously, we employ token-wise causal attention but dynamically move newly decoded tokens to the *front* of the remaining masked tokens at each step. Since the "clean" tokens at the front cannot attend to the "masked" tokens behind them, the KV cache remains valid and does not need refreshing. Crucially, during this shuffling, each token retains its original position ID. Thanks to RoPE [5], the model correctly perceives the relative positions of all tokens even when the physical sequence is reordered.
>
> 	**Motivation:** This novel strategy is necessary because existing diffusion LLMs (dLLMs) face fundamental trade-offs:
> 	- Type 1 (Global Full Attention) [1, 2]: Allows any-order generation but requires constant KV cache refreshing, making inference **slow**.
> 	- Type 2 (Block-wise Causal Attention, e.g., BD3-LM) [3, 4]: **Sacrifices global any-order flexibility** to enable KV caching.
>
> 	**From Token to Slot:** While token-level repositioning is theoretically sound, learning every possible token permutation is computationally intractable (as seen in Eso-LMs [6]). We therefore elevate this process to the slot level. By partitioning the sequence into slots, we reduce the learning complexity to a manageable slot permutation space. Motivated by our finding that the *conditional independence assumption often fails for nearby tokens* (§4.1), we define each slot as a contiguous sequence of tokens and enforce autoregressive (serial) decoding within each slot, while maintaining diffusion-based (parallel) planning between slots.
>
> 2. Detailed Workflows
>
> 	To facilitate a deeper understanding of our implementation, we provide the following explanation of the inference and training processes:
>
> 	(1) Inference Process: throughout the process, the model maintains token-wise causal attention. Each decoding step operates as a plan-and-infill cycle.
> 	- Plan: The model first predicts drafts for all masked slots in parallel. We then select a subset of high-quality slots based on specific rules (confidence scores).
> 	- Infill: The selected slots are treated as a batch and filled autoregressively to ensure local coherence.
> 	- Repositioning: After infilling, the newly decoded slots are physically moved to the front of the remaining masked slots to enable KV cache reuse. Crucially, **the position IDs for all tokens always correspond to their indices in the correct ground-truth sequence and remain invariant regardless of this repositioning.** By using these consistent position IDs with RoPE [5], the model correctly perceives relative positions even when the input buffer is reordered. The process then proceeds to the next decoding step.
> 	- Visualization: We have provided a step-by-step visualization of a complete decoding process in Appendix D.2 of our updated manuscript.
>
> 	(2) Training Process: Training is designed to enable the model to support the decoding paradigm described above. The model uses token-wise causal attention, identical to traditional ARMs. To teach the model to simultaneously learn predicting masked slots given a context of shuffled clean slots (diffusion paradigm) and filling slots sequentially (autoregressive paradigm), we process the training data as follows:
> 	- For each sample, we sample a masking ratio $t$ and mask the corresponding slots.
> 	- We then randomly permute the clean slots and place them all *before* the masked slots.
> 	- We compute an autoregressive loss for tokens within each clean slot and a diffusion loss for tokens within each masked slot.
>
> **References**
>
> [1] [Large Language Diffusion Models](https://arxiv.org/abs/2502.09992v3) NeurIPS 2025.
>
> [2] [Fast-dLLM: Training-free Acceleration of Diffusion LLM by Enabling KV Cache and Parallel Decoding](https://arxiv.org/abs/2505.22618)
>
> [3] [Block Diffusion: Interpolating Between Autoregressive and Diffusion Language Models](https://arxiv.org/abs/2503.09573) ICLR 2025.
>
> [4] [Diffusion LLMs Can Do Faster-Than-AR Inference via Discrete Diffusion Forcing](https://arxiv.org/abs/2508.09192)
>
> [5] [Roformer: Enhanced transformer with rotary position embedding](https://arxiv.org/abs/2104.09864)
>
> [6] [Esoteric Language Models](https://arxiv.org/abs/2506.01928)

---

> ### Author Response · Authors · 2025-11-21
> **Response to Reviewer wAnP (Part 2)**
>
> > Wk1 & Q1: The authors must do more to clearly and substantially differentiate ReFusion from Eso-LM. Can you please clarify what this "intractable learning objective" (lines 130-132) is and why it is intractable? What makes ReFusion different?
>
> While both our work and Eso-LMs combine Masked Diffusion Models (MDM) and Autoregressive (AR) decoding with KV cache reuse, their underlying motivations, structural designs, and training efficiencies are fundamentally different. Below, we clarify these differences from both inference and training perspectives, specifically addressing the "intractable learning objective."
>
> 1. Inference: Eso-LMs' Pipeline Strategy vs. ReFusion's Hierarchical Design
> 	- Eso-LMs: Designed to support two distinct decoding styles (MDM and AR) within one framework. It typically performs inference by first decoding a portion of tokens in parallel (MDM mode) and then switching to predicting the remaining tokens sequentially (AR mode). This is a simple concatenation of two paradigms. It does not inherently solve the issue of semantic incoherence during the parallel phase and often requires training separate models for specific MDM/AR ratios, resulting in low flexibility.
> 	- ReFusion: Introduces the concept of slots to synergistically integrate the strengths of both paradigms. We perform Inter-Slot Diffusion (parallel) for global planning and Intra-Slot Autoregressive (serial) for local infilling. This design preserves the high efficiency of parallel decoding while explicitly using AR to ensure local coherence, addressing the core weakness of standard MDMs.
>
> 2. Training
>
> 	(1) Combinatorial Complexity (Token vs. Slot): As mentioned in lines 130-132, Eso-LMs operates at the **token level**. Learning to reposition and decode tokens in any arbitrary permutation introduces a learning space of size $L!$, which is exponentially large and difficult to optimize. ReFusion partitions the sequence into fixed-size slots, reducing the permutation space from the token level to the **slot level** ($(L/k)!$). This drastic reduction makes the training objective tractable and easier to learn.
>
> 	(2) Computational and Data Efficiency:
> 	- Eso-LMs: To support its AR objective, Eso-LMs requires concatenating masked and clean sequences during AR training. This increases the attention complexity from $O(L^2)$ to roughly $O(4L^2)$. Furthermore, it cannot jointly optimize MDM and AR objectives within a single sequence, necessitating data splitting. Crucially, clean tokens serve solely as context without providing supervision. These structural limitations collectively result in highly inefficient data utilization.
> 	- ReFusion: In contrast, ReFusion employs a unified hybrid objective (Eq. 3) on a single sequence. We compute the AR loss on clean slots and the MDM loss on masked slots simultaneously within the same forward pass. This ensures **100% data utilization** (every token contributes to supervision) and maintains standard $O(L^2)$ complexity.
>
> > Wk3: The illustrative example provided in Section 5.5 is very confusing and fails to clarify the generation process. It is not clear which text segments were generated via diffusion and which were generated autoregressively.
>
> Thank you for the suggestion. To address this, we have revised and redrawn the case study visualization, which is available as Figure 5 in our updated manuscript. Tokens with orange font were generated via diffusion; tokens with black font were generated autoregressively.
>
> The revised visualization clearly shows our method's advantages:
> - High Parallelism: The figure demonstrates frequent concurrent generation of multiple slots (e.g., three separate slots generated concurrently in iteration 9).
> - Accelerated Decoding: It illustrates the significant speedup achieved by leveraging speculative decoding to retain multiple diffusion-generated tokens at once.

---

> > ### Author Response · Authors · 2025-11-21
> > **Response to Reviewer wAnP (Part 3)**
> >
> > > Wk4: The manuscript reads as heavily LLM influenced, but no disclosure is made.
> >
> > We thank the reviewer for raising this important question regarding disclosure. We confirm that the core research ideas, experimental design, and the overall content of the manuscript were developed solely by the authors. We utilized an LLM only for minor linguistic refinement and polishing of certain sentences to enhance clarity and flow. We believe this limited usage does not meet the threshold specified in the ICLR 2026 Author Guide: "if LLMs played a significant role in research ideation and/or writing to the extent that they could be regarded as a contributor." Therefore, a disclosure was not included in the initial submission.
> >
> > However, for complete transparency, we will update the manuscript to include a precise statement regarding the linguistic tools used in the final version.
> >
> > > Q2 & Q3: For the experimental benchmarks, how many tokens are being generated for each task? Is it possible that ReFusion achieves better quality simply by generating more tokens than the baselines? What are the measured latencies for ReFusion and the baseline models?
> >
> > We present the measured generated length and latency for ReFusion and the baseline models below to address the hypothesis regarding token quantity:
> >
> > ||MMLU-Pro|GSM8K|MBPP
> > |-|-|-|-
> > |Qwen3-8B|654 (20.82s)|300 (9.63s)|53 (1.74s)
> > |LLaDA-8B-Instruct|251 (13.81s)|247 (9.02s)|82 (27.65s)
> > |Dream-7B-Instruct|211 (13.23s)|223 (10.99s)|45 (36.26s)
> > |ReFusion|128 (3.25s)|148 (3.66s)|46 (1.01s)
> >
> > *Each model displays generated length (left) and measured latency in seconds (right, in parentheses).*
> >
> > The table clearly demonstrates that ReFusion's generated length is consistently shorter than or comparable to the baseline models across all tasks. This confirms that ReFusion's superior performance is not achieved by generating more tokens, but stems from the methodological superiority of the approach itself.

---

### Official Review · Reviewer_dCJA · 2025-11-02

**Soundness:** 3
**Presentation:** 2
**Contribution:** 2
**Rating:** 4
**Confidence:** 5

**Summary:**

This paper introduces a training method to achieve inter-slot parallel decoding with intra-slot autoregressive decoding for dLLMs. The core idea is that conditional independence assumption is most prone to failure for nearby tokens, and thus the authors propose to perform serial decoding within a slot, in contrast to prior method, e.g., block diffusion. The method consists of two parts. For inference,  it would first decide which blocks to decode in parallel, and then reorder the slots to do parallel decoding while still have autoregressive decoding in each slot. For training, the authors propose to adopt two objectives in training, one MDM loss for masked tokens and another AR loss for clean tokens inside the slots.

**Strengths:**

1. The authors try to challenge one common belief in current dLLM literature, that we need to perform intra-block auto-regressive decoding  instead of intra-block parallel decoding. The analysis is interesting, and it's also very interesting to see the challenge of common belief.
2. The proposed method is soundness, both for the training and the inference side.

**Weaknesses:**

My primary concern lies in the unfair, insufficient, and potentially incorrect experimental evaluations, which lead to several overstated claims in the paper.

* **Missing Comparison with Block Diffusion**

One of the key hypotheses this paper wants to show is that intra-block autoregressive decoding + inter-block parallel decoding is superior to intra-block parallel decoding + inter-block autoregressive decoding. However, no experiments are provided to substantiate this claim. A direct comparison with Block Diffusion would be necessary to validate the proposed design.  Without empirical comparison, it is unclear whether the proposed method offers any real advantage over Block Diffusion in this aspect.

* **Unfair Comparison with Prior MDMs/dLLMs**

The authors claim that Refusion overwhelmingly surpasses all prior MDMs in both performance and speed. **However, this performance gain comes mostly from a better base model.** The comparisons with previous dLLMs are not fair, as the proposed model adapts from a much stronger base model, Qwen3. While I acknowledge that the proposed method achieves the best overall results and appreciate this contribution, fair evaluation requires experiments starting from the same base model as prior works. Training from scratch may be costly, but at least one experiment comparing the proposed approach with DREAM, using the same initialization checkpoints, would be essential for a fair comparison.

* **Significant Performance Drop Compared to Qwen3, the base model of Refusion**

The adapted model exhibits an 8.5% average performance drop relative to Qwen3. This is a substantial degradation. The paper claims that the model “challenges strong ARMs,” but given such a drop in accuracy alongside a 1.34× speedup, the claim seems overstated. Besides, have you conducted experiments to evaluate whether the acceleration ratio still holds when the batch size is larger than 1?

* **Confusing Results in Table 2**

(1) For Qwen3, training with the original objective (presumably the NTP loss) results in a 10.9% performance drop, which seems unusual and needs clarification.
(2) For the GPQA benchmark (a 4-choice classification task), the reported performance is 0, which is implausible. Previous works using lm-eval-harness constrain predictions to 4 choices, meaning even random guessing should yield roughly 25% accuracy.
(3) The reported TPS (tokens per second) for LLaDA is suspiciously low. While prediction accuracy may differ, TPS should mainly depend on the model structure and sequence length. The discrepancy between Table 1 and Table 2 needs more explanation.
(4) The paper claims that the proposed model outperforms Qwen3-8B on HumanEval by ∼9 points while being 1.3× faster. However, Qwen3-8B without the authors’ training already achieves 87.8, which is actually **25.6%** higher than Refusion.

**Questions:**

See weaknesses

---

> ### Author Response · Authors · 2025-11-21
> **Response to Reviewer dCJA (Part 1)**
>
> > Wk1: Missing Comparison with Block Diffusion
>
> 1. Existing Baselines as Proxies: In our original submission, we noted that the decoding strategy of Block Diffusion is fundamentally shared by baselines like LLaDA and Fast-dLLM. Thus, their performance in Table 1 already serves as a strong proxy for the limitations of this design choice.
> 2. Direct Controlled Comparison: To provide the direct empirical validation requested, we conducted a strictly controlled head-to-head comparison by training both ReFusion and Block Diffusion [1] under identical settings, specifically using the Qwen3-8B backbone, a 120K data subset, and 10 training epochs to ensure fairness. The table below presents the performance-throughput frontier on MBPP (0-shot) across varying degrees of parallelism (Average Tokens Per Forward Pass, #TPF):
>
> |Average #TPF|1|2|3
> |-|-|-|-
> |ReFusion|56.80 (25.83)|56.00 (36.66)|47.40 (42.08)
> |Block Diffusion|51.60 (16.18)|30.20 (29.31)|16.00 (37.96)
>
> *Each model displays pass@1 (left) and throughput (TPS, right, in parentheses).*
>
> As parallelism increases, ReFusion consistently outperforms Block Diffusion in both performance and speed, maintaining high accuracy with a much gentler degradation curve. In stark contrast, Block Diffusion suffers a precipitous performance drop, underscoring the inherent methodological superiority of ReFusion.
>
> > Wk2: Unfair Comparison with Prior MDMs/dLLMs
>
> 1. Controlled Comparison with LLaDA: We selected Qwen3-8B as our backbone primarily to align with the latest research advancements in the field, rather than to gain an unfair advantage from a stronger initialization. To explicitly address the fairness concern under this experimental setting, as detailed in Section 5.4, we conducted a strictly controlled comparison by re-implementing LLaDA using the exact same Qwen3-8B backbone and identical training settings (data, epochs) as ReFusion. In this controlled environment (Table 2), ReFusion still outperformed LLaDA by **42.4%** on average while achieving **~100$\times$ higher throughput**, proving that our gains stem from the method itself, not the backbone.
> 2. Controlled Comparison with Dream (on Qwen2.5-7B): Since Dream's training code and data processing details are not open-sourced, we could not re-train it on Qwen3. Instead, following your suggestion to use the same initialization, we trained ReFusion using Dream's original checkpoint (Qwen2.5-7B). It is important to note the significant disparity in training resources:
> 	- Dream: **146.5M** samples for pre-training + 1.8M prompt-response pairs for SFT.
> 	- ReFusion: **NO** pre-training + 3.7M prompt-response pairs for SFT.
>
> 	The results are presented below:
>
> |Model|MMLU-Pro|ARC-C|GSM8K|MATH|GPQA|HumanEval|MBPP|Avg.|
> |-|-|-|-|-|-|-|-|-
> |Dream-7B-Instruct|40.05 (15.98)|88.31 (0.06)|76.42 (20.30)|46.60 (18.99)|30.36 (1.81)|56.71 (3.51)|50.40 (1.23)|55.55 (8.84)
> |ReFusion-7B|34.51 (**50.93**)|83.11 (**46.49**)|**79.38** (**41.15**)|46.38 (**45.69**)|**31.47** (**20.90**)|**69.51** (**65.57**)|**60.00** (**58.49**)|**57.77** (**47.03**)
>
> *Each model displays accuracy/pass@1 (left) and throughput (TPS, right, in parentheses).*
>
> ReFusion achieves a **5.32$\times$ speedup** and a **2.22% average performance gain** over Dream. ReFusion significantly outperforms Dream on reasoning and coding tasks (GSM8K, HumanEval, MBPP). We observe lower performance on knowledge-intensive tasks (MMLU-Pro, ARC-C), which is expected given that ReFusion skipped the massive pre-training stage (146.5M samples) that Dream utilized to inject knowledge.
>
> These results confirm that ReFusion's advantages, specifically the 'plan-and-infill' mechanism and full KV cache reuse, are robust across different base models.

---

> ### Author Response · Authors · 2025-11-21
> **Response to Reviewer dCJA (Part 2)**
>
> > Wk3: Significant Performance Drop Compared to Qwen3, the base model of Refusion. Besides, have you conducted experiments to evaluate whether the acceleration ratio still holds when the batch size is larger than 1?
>
> 1. Clarifying the Performance Drop: This is primarily attributed to two factors:
> 	- Data Quality: Our fine-tuning relies on open-source datasets, which naturally differ in quality and distribution from the proprietary data used to train the Qwen3 base model.
> 	- Objective Shift: Adapting a model from next-token prediction (AR) to diffusion (MDM) involves a significant paradigm shift. Even existing strong MDMs initialized from strong ARMs experience this; for instance, Dream (initialized from Qwen2.5-7B) shows performance drops of 1.60% on Math and 3.10% on Code relative to its base [2], despite using significantly more compute and data than ReFusion.
> 2. Clarifying the "Challenge" Claim: Our primary goal is to demonstrate ReFusion's methodological superiority as an MDM rather than to surpass the absolute performance ceiling of all architectures. ReFusion is among the first MDMs to successfully surpass ARMs in speed (e.g., **1.4$\times$ speedup**) while simultaneously matching or exceeding their performance on complex tasks like GSM8K and MBPP. This represents a significant milestone in making MDMs a viable competitor to ARMs.
> 3. Acceleration Ratio with Larger Batch Sizes: Following your suggestion, we measured the model's forward pass time on a single A100 GPU across varying batch sizes. The table below displays the throughput (TPS) of different models across varying batch sizes:
>
> |Batch Size|1|2|4|8|16|32|64
> |-|-|-|-|-|-|-|-
> |Qwen3-8B|32.42|58.66|110.97|215.77|397.21|664.69|839.07
> |Block Diffusion|16.18|35.18|67.79|135.15|253.54|369.44|420.86
> |ReFusion|45.34|88.78|164.22|292.78|436.77|561.09|624.47
>
> - Vs. MDM Baseline: ReFusion consistently outperforms Block Diffusion across all batch sizes.
> - Vs. ARM: ReFusion maintains its speed advantage over Qwen3 for batch sizes up to 16.
> - Compute vs. Memory Bound: As batch size increases beyond 32, ARMs overtake ReFusion. This is a known characteristic of MDMs: they require more computation per step (denoising) than ARMs. At low batch sizes, inference is memory-bound, allowing ReFusion's parallel decoding to shine. As batch size increases, the process shifts to a compute-bound regime, where the higher FLOPs count of MDMs becomes the bottleneck. We plan to address this limitation in future work.
>
> > Wk4: Confusing Results in Table 2
>
> > Wk4.1: For Qwen3, training with the original objective (presumably the NTP loss) results in a 10.9% performance drop, which seems unusual and needs clarification.
>
> Qwen3-8B has undergone extensive post-training and is already well-optimized across various tasks. Therefore, further training it with open-source data, which may be of relatively lower quality compared to the model's original proprietary training corpus, inevitably leads to a degradation in performance.
>
> > Wk4.2: For the GPQA benchmark (a 4-choice classification task), the reported performance is 0, which is implausible. Previous works using lm-eval-harness constrain predictions to 4 choices, meaning even random guessing should yield roughly 25% accuracy.
>
> We would like to clarify that for the GPQA benchmark, we adhered to the evaluation protocols used by LLaDA [3] and Dream [2]. Specifically, we evaluated the instruct models using conditional generation via the `gpqa_main_generative_n_shot` task in `lm-evaluation-harness`.
>
> Unlike the likelihood-based multiple-choice evaluation (which constrains predictions to A/B/C/D and guarantees ~25% random baseline), this generative evaluation method relies on extracting the final answer from the model's free-form text response. Consequently, if a model fails to generate a recognizable answer format, the score can fall below 25%. The 0% score for LLaDA in Table 2 results from its poor instruction-following capabilities in that specific setting, where it failed to generate responses that could be successfully extracted by the evaluator.

---

> > ### Author Response · Authors · 2025-11-21
> > **Response to Reviewer dCJA (Part 3)**
> >
> > > Wk4.3: The reported TPS (tokens per second) for LLaDA is suspiciously low. While prediction accuracy may differ, TPS should mainly depend on the model structure and sequence length. The discrepancy between Table 1 and Table 2 needs more explanation.
> >
> > We clarify that the discrepancy in TPS between Table 1 and Table 2 arises from substantial differences in the underlying model architectures used in the two experiments.
> > - In Table 1: The results are reported directly from the original LLaDA paper [3]. That model consists of **32 layers** and has a vocabulary size of **126,464**.
> > - In Table 2: To ensure a strictly fair comparison with ReFusion (which uses Qwen3-8B), we re-implemented and trained LLaDA using Qwen3-8B as the initialization checkpoint following the method in [3]. This Qwen3-based model is deeper and wider, consisting of **36 layers** with a significantly larger vocabulary size of **151,936**.
> >
> > The increased model depth and larger vocabulary size in the Qwen3-based implementation (Table 2) introduce higher computational overhead, naturally resulting in a lower TPS compared to the original LLaDA model in Table 1.
> >
> > > Wk4.4: The paper claims that the proposed model outperforms Qwen3-8B on HumanEval by ∼9 points while being 1.3× faster. However, Qwen3-8B without the authors’ training already achieves 87.8, which is actually 25.6% higher than Refusion.
> >
> > There may be some misunderstandings due to the ambiguous labeling. To clarify, it would be more accurate to label the "Qwen3-8B" entry in Table 2 as "ARM". The specific purpose of Table 2 is to present a controlled comparison that isolates the impact of the training method by eliminating variables such as initialization checkpoints and data quality. In this experiment, the model labeled "Qwen3-8B" refers to the backbone trained on the same 120K data subset using the standard autoregressive objective, rather than the fully post-trained, off-the-shelf Qwen3-8B (which indeed scores 87.8 as reported in Table 1).
> >
> > Our results demonstrate that under these identical, controlled training conditions, ReFusion outperforms the standard ARM baseline by ~9 points on HumanEval while being 1.3$\times$ faster, and consistently surpasses the MDM baseline (LLaDA) in both speed and performance. We will update the label in the final revision to prevent further confusion.
> >
> > **References**
> >
> > [1] [Block Diffusion: Interpolating Between Autoregressive and Diffusion Language Models](https://arxiv.org/abs/2503.09573) ICLR 2025.
> >
> > [2] [Dream 7B: Diffusion Large Language Models](https://arxiv.org/abs/2508.15487)
> >
> > [3] [Large Language Diffusion Models](https://arxiv.org/abs/2502.09992v3) NeurIPS 2025.

---

### Official Review · Reviewer_VXks · 2025-11-02

**Soundness:** 3
**Presentation:** 2
**Contribution:** 3
**Rating:** 6
**Confidence:** 4

**Summary:**

This paper proposes ReFusion, a masked-diffusion LLM that performs plan-and-infill decoding at a slot level: a diffusion planner selects weakly dependent fixed-length spans (“slots”), and an AR stage that fills multiple slots in parallel. By reordering into a causal sequence during infill, the method reuses exact KV cache, narrowing the cache-efficiency gap between diffusion LMs and AR models.

Across various benchmarks and two dLLM architectures (LLaDA, Dream), ReFusion reports consistent >10× speedup over prior MDMs, and about 1.4x faster than SOTA AR models.

**Strengths:**

1. The slot abstraction plus causal infill provides an intuitive route to exact KV-cache reuse, reducing efficiency gap with AR decoding.

2. The paper provides comparison with competitive baselines, and it shows consistent wins on most tasks with both LLaDA and Dream across many tasks, supporting generality.

3. The paper probes slot thresholds and provides qualitative evidence that aligns with the design intuition.

**Weaknesses:**

1. ReFusion introduces extra data preparation and training cost, adding pipeline complexity to realize its gains.

2. While ReFusion is much faster than prior MDMs, its throughput versus AR models is not significant better, and ReFusion may not be orthogonal to existing tricks.

**Questions:**

1. The authors mention 3.7M samples; approximately how many tokens does this correspond to, and how long does training take on 16x8 H20 GPUs (total GPU-hours)?

2. How orthogonal is ReFusion to other dLLM acceleration techniques (e.g. confidence-aware parallelism)? Could the slot planner or infill step compose with the techniques, and if so, what additive speed/quality gains should we expect?

3. Do the authors have preliminary results on how does scaling the training data beyond 3.7M samples shift the speed–quality frontier?

4. In Table 1, why is D2F significantly worse on ARC-C than its performance on other tasks? any intuitions?

---

> ### Author Response · Authors · 2025-11-21
> **Response to Reviewer VXks (Part 1)**
>
> > Wk1: ReFusion introduces extra data preparation and training cost, adding pipeline complexity to realize its gains.
>
> We respectfully submit that there might be a misunderstanding regarding the training overhead. We would like to clarify that ReFusion actually **reduces** the overall training burden compared to existing MDM baselines.
>
> 1. Identical Computational Cost per Step: Given identical network topology and input tensor dimensions, the theoretical FLOPs for a single forward pass of ReFusion are **exactly the same** as those of traditional ARMs and MDMs. Therefore, our method introduces no extra computational overhead during the training steps themselves.
> 2. Significantly Lower Data and Training Requirements: Any new model requires data for training. However, unlike baseline MDMs which require massive, resource-intensive **pre-training** on hundreds of millions of samples, ReFusion **skips the pre-training stage entirely**. We only require lightweight supervised fine-tuning to adapt the model to the MDM paradigm. The comparison of training costs is detailed in the table below:
>
> ||Pre-training|Supervised Fine-Tuning
> |-|-|-
> |LLaDA|561.5 million samples|4.5 million prompt-response pairs, 3 epochs
> |Dream|146.5 million samples|1.8 million prompt-response pairs, 3 epochs
> |ReFusion|N/A|3.7 million prompt-response pairs, 4 epochs
>
> As shown, ReFusion achieves superior performance with a fraction of the data and computational resources, effectively lowering the barrier to entry rather than adding complexity.
>
> > Wk2: While ReFusion is much faster than prior MDMs, its throughput versus AR models is not significant better, and ReFusion may not be orthogonal to existing tricks.
>
> 1. Comparison with ARMs and MDMs: ReFusion is a MDM that operates by progressively denoising a fully masked sequence; therefore, our primary comparison naturally focuses on existing MDMs, against which ReFusion demonstrates overwhelming advantages in both speed and accuracy.
>
> 	Regarding the comparison with ARMs:
> 	- Breaking the Trade-off: Prior MDMs generally lagged behind ARMs in both metrics; attempts to increase their speed typically resulted in catastrophic performance drops.
> 	- A New Frontier: ReFusion is among the first MDMs to successfully surpass ARMs in speed (e.g., **1.4$\times$ speedup**) while simultaneously matching or exceeding their performance on complex tasks like GSM8K and MBPP. This represents a significant milestone in making MDMs a viable competitor to ARMs.
>
> 2. Relationship with Existing Tricks: The relationship between ReFusion and existing optimization strategies can be categorized into three types:
> 	- Incompatible (Redundant): Many tricks were specifically designed to enable KV caching or variable-length generation for MDMs with bidirectional attention (e.g., Fast-dLLM [1], DAEDAL [2]). Since ReFusion natively supports full KV cache reuse and efficient variable-length generation through its architectural design, these "workaround" tricks are rendered unnecessary and are thus incompatible.
> 	- Compatible (Adopted): We are fully compatible with and have integrated general-purpose optimizations that are decoupled from the specific mechanism. For instance, ReFusion already employs the semi-autoregressive remasking strategy [1, 3, 4] and confidence-aware parallel decoding strategy [1, 5] to enhance efficiency.
> 	- Advanced (Integrated): We also leverage advanced techniques that are being explored in MDMs but originated elsewhere. For example, our "plan-and-infill" mechanism naturally incorporates speculative decoding [6], using the diffusion step to propose draft slots that are subsequently verified and completed.
>
> > Q1: The authors mention 3.7M samples; approximately how many tokens does this correspond to, and how long does training take on 16x8 H20 GPUs (total GPU-hours)?
>
> The 3.7M samples used in our training correspond to approximately 1.22 billion (B) tokens. The total training cost was approximately 10.68K H20 GPU-hours.
>
> **References**
>
> [1] [Fast-dLLM: Training-free Acceleration of Diffusion LLM by Enabling KV Cache and Parallel Decoding](https://arxiv.org/abs/2505.22618)
>
> [2] [Beyond Fixed: Training-Free Variable-Length Denoising for Diffusion Large Language Models](https://arxiv.org/abs/2508.00819)
>
> [3] [Block Diffusion: Interpolating Between Autoregressive and Diffusion Language Models](https://arxiv.org/abs/2503.09573) ICLR 2025.
>
> [4] [Large Language Diffusion Models](https://arxiv.org/abs/2502.09992v3) NeurIPS 2025.
>
> [5] [Diffusion LLMs Can Do Faster-Than-AR Inference via Discrete Diffusion Forcing](https://arxiv.org/abs/2508.09192)
>
> [6] [Fast inference from transformers via speculative decoding](https://arxiv.org/abs/2211.17192) ICML 2023.

---

> > ### Author Response · Authors · 2025-11-21
> > **Response to Reviewer VXks (Part 2)**
> >
> > > Q2: How orthogonal is ReFusion to other dLLM acceleration techniques (e.g. confidence-aware parallelism)? Could the slot planner or infill step compose with the techniques, and if so, what additive speed/quality gains should we expect?
> >
> > We would like to clarify that ReFusion already integrates confidence-aware parallelism into its inference strategy. Specifically:
> > - In the Diffusion-based Slot Planning phase: We select only those slots whose certainty scores exceed the threshold $\tau$.
> > - In the Autoregressive Slot Infilling phase: When applying speculative decoding, we retain the longest prefix of the concatenated draft where the probabilities of all tokens exceed the threshold $\tau$.
> >
> > To demonstrate the additive gains, the table below compares the performance and throughput (TPS) of ReFusion with and without confidence-aware parallelism:
> > ||GSM8K (0-shot)|MBPP (0-shot)|MMLU-Pro (0-shot)
> > |-|-|-|-
> > |w/ confidence-aware parallelism|85.60 (40.29)|66.60 (45.34)|45.39 (39.38)
> > |w/o confidence-aware parallelism|85.75 (28.95)|68.20 (25.19)|46.15 (28.93)
> >
> > *Each method displays accuracy/pass@1 (left) and throughput (TPS, right, in parentheses).*
> >
> > The results indicate that incorporating confidence-aware parallelism yields a ~1.5$\times$ speedup across tasks with only a negligible performance drop (0.84% on average).
> >
> > > Q3: Do the authors have preliminary results on how does scaling the training data beyond 3.7M samples shift the speed–quality frontier?
> >
> > We thank the reviewer for this suggestion. To investigate the scaling potential, we collected additional data (slightly lower quality) to expand our training set from the original 3.7M samples to 14M samples. The comparative results are presented below:
> >
> > |Data Size|GSM8K (0-shot)|MBPP (0-shot)|MMLU-Pro (0-shot)
> > |-|-|-|-
> > |120K|73.16 (32.54)|55.60 (31.01)|36.84 (29.64)
> > |1M|82.18 (36.37)|62.40 (34.72)|38.81 (29.00)
> > |2M|83.40 (36.85)|59.20 (34.96)|40.73 (30.76)
> > |3.7M|81.05 (36.78)|61.20 (36.59)|39.93 (31.02)
> > |14M|81.12 (37.58)|62.80 (39.63)|45.31 (37.05)
> >
> > *Each model displays accuracy/pass@1 (left) and throughput (TPS, right, in parentheses).*
> >
> > The results clearly indicate that scaling the training data beyond 3.7M shifts the speed-quality frontier further outward. This suggests that ReFusion possesses significant scalability, with the potential to unlock even greater performance gains given higher-quality data and increased computational resources.
> >
> > > Q4: In Table 1, why is D2F significantly worse on ARC-C than its performance on other tasks? any intuitions?
> >
> > The lower throughput on ARC-C is primarily due to the mismatch between D2F's fixed block size and the short generation length required by the task.
> >
> > In Table 1, throughput (TPS) is calculated as Valid Generated Tokens / Latency. Crucially, the numerator counts only the effective tokens generated before the End-of-Sequence (EOS) token. Following the official implementation of D2F, the block size is fixed at 32. This means the model incurs the computational cost of generating at least 32 tokens, regardless of the actual answer length. ARC-C is a multiple-choice task that typically requires very short responses (e.g., just the option letter). In our experiments, the average valid generation lengths for LLaDA w/ D2F and Dream w/ D2F were only 1 token and 2.57 tokens, respectively. Although the model physically generates a full block of 32 tokens (incurring high latency), only a tiny fraction (1~3 tokens) contributes to the TPS numerator. This "padding overhead" drastically reduces the calculated throughput compared to tasks with longer responses (like GSM8K) where the valid tokens fill the blocks more efficiently.

---

### Official Review · Reviewer_ZGiW · 2025-11-03

**Soundness:** 3
**Presentation:** 2
**Contribution:** 2
**Rating:** 4
**Confidence:** 3

**Summary:**

This paper focuses on two intrinsic drawbacks of masked diffusion models (MDMs): the infeasibility of KV caching and missing token dependencies.
To alleviate these issues, the authors propose to combine the AR and MDM paradigm, called the ReFusion Model.
Although this idea is not very novel, the authors discuss in detail the practical design of the ``plan-and-infilling'' process.
Extensive experiments demonstrate the generation speedup benefiting from the KV caching and the improved accuracy gain.

**Strengths:**

- The authors propose a practical ``plan-and-infilling'' process that works fairly well, although the basic idea of ReFusion might not be very novel.
- The pilot study in Section 4.1 is very interesting and clearly exhibits how the distance affects the correlation.
- The experiments, including the ablation studies, provide a very comprehensive understanding of how ReFusion works.

**Weaknesses:**

- Similar ideas have been discussed in previous works. E.g., BD3-LM (https://arxiv.org/pdf/2503.09573) utilizes the block diffusion, EDLM (https://arxiv.org/abs/2410.21357) utilize AR models to model the correlations. The authors should clearly state the differences and their unique contribution.
- I find the two-step inference method in Section 4.2 somewhat obscure. I suggest the authors reorganize the desciption into mathematical equations and add more details (e.g., how to perform positional encoding for $S_t^{clean}$ and $S_t^{mask}$).
- The authors should discuss more detailedly how why the use the slot's first position to compute the certainty score. For example, the mean of the probs on all positions in a slot seems to be a more appropriate method.
- As an intrinsic drawback of the ReFusion method, there are many critical hyper-parameters to be tuned. e.g, $k$ and $\tau$ and $\lambda$. These parameters may depends on the data sets, and can affect the broader application of ReFusion.
- Why is the weight $1/t$ missing from MDM objective in equation (2)?

**Questions:**

Please see the Weaknesses.

---

> ### Author Response · Authors · 2025-11-21
> **Response to Reviewer ZGiW (Part 1)**
>
> > Wk1: Similar ideas have been discussed in previous works. E.g., BD3-LM utilizes the block diffusion, EDLM utilize AR models to model the correlations. The authors should clearly state the differences and their unique contribution.
>
> While ReFusion shares high-level concepts like "partitioning" or "combining AR and Diffusion" with prior methods, our **motivation and mechanism** are fundamentally different.
>
> 1. To clearly differentiate our work, we first summarize the core motivation and unique mechanism of ReFusion:
> 	Existing diffusion large language models (dLLMs) generally fall into two categories:
> 	- Type 1 (Global Full Attention) [1, 2]: Allows any-order decoding (every masked token attends to all clean tokens), but requires constant KV cache refreshing, making inference **slow**.
> 	- Type 2 (Block-wise Causal Attention, e.g., BD3-LM) [3, 4]: Enforces serial generation between blocks to enable KV caching, but allows any-order decoding within blocks. While this improves efficiency, it **sacrifices global any-order flexibility**, limiting the model's potential to discover optimal generation paths beyond a fixed left-to-right order.
>
> 	**Our Contribution:** To achieve **both high efficiency (via KV cache reuse) and global flexibility (any-order decoding)**, ReFusion introduces a novel strategy:
> 	- Mechanism: We maintain any-order flexibility by not constraining the model’s generation order. To support KV cache reuse simultaneously, we employ token-wise causal attention but dynamically move newly decoded tokens to the *front* of the remaining masked tokens at each step. Since the "clean" tokens at the front cannot attend to the "masked" tokens behind them, the KV cache remains valid and does not need refreshing. Crucially, during this shuffling, each token retains its original position ID. Thanks to RoPE [5], the model correctly perceives the relative positions of all tokens even when the physical sequence is reordered.
> 	- From Token to Slot: While token-level repositioning is theoretically sound, learning every possible token permutation is computationally intractable (as seen in Eso-LMs [6]). We therefore elevate this process to the slot level. By partitioning the sequence into slots, we reduce the learning complexity to a manageable slot permutation space. Motivated by our finding that the *conditional independence assumption often fails for nearby tokens* (§4.1), we define each slot as a contiguous sequence of tokens and enforce autoregressive (serial) decoding within each slot, while maintaining diffusion-based (parallel) planning between slots.
>
> 	In summary, ReFusion is not merely an ensemble of existing techniques but a novel framework enabling **Inter-Slot Any-Order + Intra-Slot Autoregressive decoding with precise KV Cache reuse.**
>
> 2. Comparison with BD3-LM: Although both methods partition the sequence, the definitions and goals of 'blocks' vs. 'slots' are fundamentally different.
> 	- BD3-LM: Uses "blocks" primarily to enable KV caching. It enforces **Inter-Block Serial** (rigid left-to-right) and **Intra-Block Parallel** decoding. This sacrifices global flexibility for cache reuse.
> 	- ReFusion: Uses "slots" primarily to reduce training complexity. We enable **Inter-Slot Parallel** (flexible any-order) and **Intra-Slot Serial** decoding. This preserves global flexibility and enables full KV cache reuse, while solving the local incoherence issue common in diffusion models.
>
> 3. Comparison with EDLM: Both methods aim to model correlations to improve generation quality, but the solutions differ significantly:
> 	- EDLM: Relies on an **external** autoregressive model (ARM) to verify the sequence generated at each step. This introduces the computational overhead of maintaining and querying a second model.
> 	- ReFusion: Achieves coherence through a **unified single model**. By simply training with a hybrid objective (AR loss for intra-slot, diffusion loss for inter-slot), ReFusion learns to generate coherent sequences serially within slots without needing an auxiliary ARM.
>
> **References**
>
> [1] [Large Language Diffusion Models](https://arxiv.org/abs/2502.09992v3) NeurIPS 2025.
>
> [2] [Fast-dLLM: Training-free Acceleration of Diffusion LLM by Enabling KV Cache and Parallel Decoding](https://arxiv.org/abs/2505.22618)
>
> [3] [Block Diffusion: Interpolating Between Autoregressive and Diffusion Language Models](https://arxiv.org/abs/2503.09573) ICLR 2025.
>
> [4] [Diffusion LLMs Can Do Faster-Than-AR Inference via Discrete Diffusion Forcing](https://arxiv.org/abs/2508.09192)
>
> [5] [Roformer: Enhanced transformer with rotary position embedding](https://arxiv.org/abs/2104.09864)
>
> [6] [Esoteric Language Models](https://arxiv.org/abs/2506.01928)

---

> > ### Author Response · Authors · 2025-11-21
> > **Response to Reviewer ZGiW (Part 2)**
> >
> > > Wk2: I find the two-step inference method in Section 4.2 somewhat obscure. I suggest the authors reorganize the desciption into mathematical equations and add more details (e.g., how to perform positional encoding for $S_t^{clean}$ and $S_t^{mask}$).
> >
> > We thank the reviewer for the suggestion. We would like to clarify that the detailed mathematical formalization of the inference method is provided in Appendix A.2. To address your concern and further clarify the process, our inference mechanism can be summarized as follows:
> >
> > 1. The "Plan-and-Infill" Paradigm: The model maintains token-wise causal attention throughout the entire process. Each decoding iteration operates as a "plan-and-infill" cycle:
> > 	- Plan: The model first predicts drafts for all masked slots in parallel. We then select a subset of high-quality slots based on specific rules (confidence scores).
> > 	- Infill: The selected slots are treated as a batch and filled autoregressively to ensure local coherence.
> >
> > 2. Positional Encoding and Reordering: After infilling, the newly decoded slots are physically moved to the front of the remaining masked slots to enable KV cache reuse. Crucially, **the position IDs for all tokens always correspond to their indices in the correct ground-truth sequence and remain invariant regardless of this repositioning.** By using these consistent position IDs with RoPE [5], the model correctly perceives relative positions even when the input buffer is reordered.
> >
> > For a more intuitive understanding, we have provided a step-by-step visualization of a complete decoding process in Appendix D.2 of our updated manuscript.
> >
> > > Wk3: The authors should discuss more detailedly how why the use the slot's first position to compute the certainty score. For example, the mean of the probs on all positions in a slot seems to be a more appropriate method.
> >
> > We thank the reviewer for this insightful suggestion. Our decision to use the probability of the first token as the certainty score is motivated by the specific dynamics of our "plan-and-infill" inference mechanism. In the Planning phase, the primary goal is to identify valid starting points for slots. While the diffusion model drafts the entire slot, the subsequent autoregressive infilling phase is responsible for generating the subsequent tokens conditioned on the first one. Therefore, the confidence of the first token serves as a sufficient proxy indicator for whether a slot should be initiated. However, we agree that the mean probability is also a sound alternative. Following your suggestion, we compared the two methods (see table below). The results are highly comparable to Table 1, indicating that both metrics select a significantly overlapping set of slots for parallel decoding.
> >
> > |Method|MMLU-Pro|ARC-C|GSM8K|MATH|GPQA|HumanEval|MBPP
> > |-|-|-|-|-|-|-|-
> > |Prob. of First Token|45.39|89.68|85.60|56.06|35.04|75.61|66.60
> > |Mean Prob. of Slot|45.18|89.68|85.14|56.14|33.71|77.44|67.80
> >
> > > Wk4: As an intrinsic drawback of the ReFusion method, there are many critical hyper-parameters to be tuned. e.g, $k$ and $\tau$ and $\lambda$. These parameters may depends on the data sets, and can affect the broader application of ReFusion.
> >
> > We would like to clarify that these hyperparameters are either standard in the field of dLLMs or require minimal tuning in practice:
> > - $k$: corresponds to the slot/block size widely used in other partitioning-based methods [1, 2, 3, 4].
> > - $\tau$: a common parameter to control the confidence-based parallelism in dynamic decoding frameworks [2, 4, 7].
> > - $\lambda$: balances different loss types; notably, it was set to **1** across **all** our experiments and required no tuning.
> >
> > Moreover, our sensitivity analysis in Section 5.3 demonstrates that ReFusion is highly robust to these settings. We identified a wide "sweet spot" where performance consistently surpasses baselines, recommending $\tau \in [0.5, 0.9]$ and $k \in \{16, 32\}$. Finally, we note that the reliance on inference hyperparameters is not unique to ReFusion. Standard ARMs also depend on tuning parameters such as temperature, top-$p$, and top-$k$ to achieve optimal performance across different tasks. The hyperparameter complexity of ReFusion is thus comparable to established paradigms.
> >
> > **References**
> >
> > [1] [Large Language Diffusion Models](https://arxiv.org/abs/2502.09992v3) NeurIPS 2025.
> >
> > [2] [Fast-dLLM: Training-free Acceleration of Diffusion LLM by Enabling KV Cache and Parallel Decoding](https://arxiv.org/abs/2505.22618)
> >
> > [3] [Block Diffusion: Interpolating Between Autoregressive and Diffusion Language Models](https://arxiv.org/abs/2503.09573) ICLR 2025.
> >
> > [4] [Diffusion LLMs Can Do Faster-Than-AR Inference via Discrete Diffusion Forcing](https://arxiv.org/abs/2508.09192)
> >
> > [5] [Roformer: Enhanced transformer with rotary position embedding](https://arxiv.org/abs/2104.09864)
> >
> > [7] [Fast-dLLM v2: Efficient Block-Diffusion LLM](https://arxiv.org/abs/2509.26328)

---

> > > ### Author Response · Authors · 2025-11-21
> > > **Response to Reviewer ZGiW (Part 3)**
> > >
> > > > Wk5: Why is the weight $1/t$ missing from MDM objective in equation (2)?
> > >
> > > We would like to clarify that the $1/t$ weight is implicitly included in the normalization factor of Equation (2). Specifically, let $K$ denote the total number of slots in the response. The number of masked slots at timestep $t$ is $|\boldsymbol{S}^{\text{masked}}_t| = t \cdot K$. Consequently, the normalization term can be decomposed as follows:
> > >
> > > $$\frac{1}{k \cdot |\boldsymbol{S}^{\text{masked}}_t|} = \frac{1}{k \cdot (t \cdot K)} = \frac{1}{k \cdot K} \cdot \frac{1}{t}$$
> > >
> > > Thus, Equation (2) can be equivalently rewritten to explicitly show the $1/t$:
> > >
> > > $$ \mathcal{L} _ {MDM} = \frac{1}{k \cdot K} \cdot -\mathbb{E} _ {\begin{subarray}{l} (p _ 0,r _ 0)\sim\mathcal{D} \\ t\sim U(0,1) \end{subarray}}\left[\frac{1}{t}\sum _ {i=1} ^ {|\boldsymbol{S} ^ {\text{masked}} _ t|}\sum _ {j=1} ^ {k}\log P _ \theta(v _ 0 ^ {i,j} \mid p _ 0, \boldsymbol{S} ^ {\text{clean}} _ {t}, \boldsymbol{S} ^ {\text{masked}} _ {t,\leqslant(i,j)})\right] $$
> > >
> > > This formulation calculates the average MDM loss per token. We will add this derivation to our revision to avoid future confusion.

---

### Author Response · Authors · 2025-12-02
**Rebuttal Summary and Key Updates**

Dear Area Chairs,

We sincerely appreciate your time and additional effort in managing the review process under these unprecedented circumstances. To assist your evaluation, we provide a high-level summary of how we have addressed the reviewers’ concerns through clarifications and significant new experiments.

**Core Contribution: A New Paradigm for MDMs**

MDMs traditionally force a trade-off between the efficiency of KV Caching and the flexibility of any-order decoding. ReFusion introduces a novel slot-based “plan-and-infill” paradigm that resolves this core conflict. Our key innovation is twofold: (1) It unifies a causal attention mechanism with global, any-order slot generation, enabling full KV cache reuse without sacrificing flexibility; (2) It simplifies the learning objective from an intractable token-combination space to a manageable slot-permutation space, boosting learning efficiency.

The practical result is a new state-of-the-art for MDMs. Experimentally, ReFusion not only surpasses all prior MDMs in both speed and performance but also bridges the gap to strong ARMs, sometimes even outperforming them on both metrics simultaneously. We respectfully request that you consider this fundamental methodological leap and its significant practical breakthroughs in your final decision.

### **Major Concerns & Improvements in Revised Version**

1. Novelty and Methodological Distinctness (Reviewer ZGiW: Wk1, Wk2; Reviewer wAnP: Wk1, Wk2, Q1)

	Reviewers requested clearer differentiation from prior art like BD3-LMs and Eso-LMs. We clarified that these methods make critical compromises (e.g., sacrificing global flexibility or introducing intractable training objectives). In contrast, ReFusion's slot-based framework is specifically designed to resolve the "KV cache vs. Flexibility" trade-off without compromise. To substantiate this, we have provided a detailed mathematical formalization (Appendix A.2) and added a step-by-step visualization of our unique decoding process (Appendix D.2).

2. Comparison with Existing MDMs and ARMs (Reviewer VXks: Wk2; Reviewer dCJA: Wk1, Wk2, Wk3)

	While reviewers acknowledged our superiority over MDMs, they questioned the magnitude of gains against strong ARMs. We clarify that our primary goal is to demonstrate ReFusion's methodological superiority as an MDM rather than solely to surpass the absolute performance ceiling of all architectures. Nevertheless, ReFusion represents a milestone: it is among the first MDMs to surpass ARMs in speed while matching performance on complex tasks (e.g., GSM8K, MBPP). To strengthen our empirical validation, we added a performance-throughput frontier comparison with Block Diffusion in Section 5.5 and a controlled comparison against Dream using the same initialization checkpoint in Section 5.4. Both experiments further confirm our method's advantages in speed and performance.

3. Ablations of Methodological Components (Reviewer ZGiW: Wk3; Reviewer VXks: Q2)

	Reviewer ZGiW questioned the motivation behind our slot certainty scoring method and suggested alternatives and we provided a detailed explanation and additional experimental results in Appendix C.2. Reviewer VXks inquired about the ablation of the confidence-aware parallelism technique and we added experiments in Appendix C.5 demonstrating that this technique achieves approximately 1.5$\times$ acceleration.

4. Scalability (Reviewer VXks: Q3)

	Regarding the question of how ReFusion's speed-quality frontier shifts with data scaling, we extended our training data to 14M samples in Appendix C.3. The results demonstrate significant scalability, indicating the potential to unlock even greater performance gains given higher-quality data and increased computational resources.

5. Enhanced Clarity

	- Hyperparameters (Reviewer ZGiW: Wk4): We clarified in Section 5.3 that ReFusion’s hyperparameters are either standard within the MDM field or fixed values that require no tuning.
	- Experimental Details: We addressed specific queries regarding experimental procedures and results in the revised paper: Section 5 & Appendix B.1 (Reviewer VXks: Wk1 & Q1 & Q4); Section 5.4 (Reviewer dCJA: Wk4); Section 5.6 & Appendix C.1 (Reviewer wAnP: Wk3 & Q2 & Q3).
	- Presentation: We refined the phrasing in Footnote on Page 6 (Reviewer ZGiW: Wk5) and Appendix F (Reviewer wAnP: Wk4) to improve clarity.

We appreciate the recognition of the novelty and significance of our method in challenging the prevailing paradigms of existing MDMs (VXks, dCJA, wAnP), the comprehensive experimental results demonstrating superior performance-throughput trade-offs (ZGiW, VXks, wAnP), and the insightful analysis of local token dependency (ZGiW, dCJA, wAnP). We believe our responses have effectively addressed the relevant concerns raised by the reviewers.

We hope this summary assists you in reviewing our submission more comfortably and efficiently.

Thank you again for your time and consideration.

Sincerely,

The Authors

---

### Meta-Review · Area_Chair_xF8n · 2026-01-07

**Summary:**

Based on the reviews and the authors' rebuttal, I recommend accepting this paper. While reviewers initially raised valid concerns regarding the fairness of comparisons across different model backbones, the differentiation from prior block-based diffusion models, and the clarity of the KV cache reuse mechanism, the authors effectively addressed these issues through rigorous controlled experiments and detailed methodological clarifications. The novel slot-based "plan-and-infill" strategy, which uniquely enables efficient parallel decoding with full KV cache reuse, combined with extensive empirical validation demonstrating superior performance-throughput trade-offs against strong baselines, makes this a significant contribution to the field.

**Reviewer Concerns:**

Addressed:
1. Fairness of Comparison (Reviewer dCJA): The critical concern that performance gains relied on the Qwen3 backbone was refuted. The authors provided controlled experiments retraining baselines on the exact same backbone, where the proposed method still maintained superior performance.
2. Novelty & Differentiation (Reviewers ZGiW, wAnP): The rebuttal clearly distinguished ReFusion from BD3-LM and Eso-LM.
3. Mechanism Clarity (Reviewer wAnP): The confusion regarding how KV cache is reused was resolved by explaining the position-ID invariant reordering strategy and adding step-by-step visualizations.
4. Scalability (Reviewer VXks): New experiments demonstrated positive scaling laws for both speed and quality up to 14M samples.

**Reviewer Scores:**

1. Reviewer ZGiW: 4 -> 6
2. Reviewer VXks: 6 -> 6
3. Reviewer dCJA: 4 -> 6
4. Reviewer wAnP: 4 -> 6

---

### Decision · Program_Chairs · 2026-01-26

Accept (Poster)